# Targeting Cancer Stem Cells by Dietary Agents: An Important Therapeutic Strategy against Human Malignancies

**DOI:** 10.3390/ijms222111669

**Published:** 2021-10-28

**Authors:** Mahshid Deldar Abad Paskeh, Shafagh Asadi, Amirhossein Zabolian, Hossein Saleki, Mohammad Amin Khoshbakht, Sina Sabet, Mohamad Javad Naghdi, Mehrdad Hashemi, Kiavash Hushmandi, Milad Ashrafizadeh, Sepideh Mirzaei, Ali Zarrabi, Gautam Sethi

**Affiliations:** 1Department of Genetics, Faculty of Advanced Science and Technology, Tehran Medical Sciences, Islamic Azad University, Tehran 1916893813, Iran; deldar_mahshid@yahoo.com (M.D.A.P.); mhashemi@iautmu.ac.ir (M.H.); 2Farhikhtegan Medical Convergence Sciences Research Center, Farhikhtegan Hospital Tehran Medical Sciences, Islamic Azad University, Tehran 1916893813, Iran; 3Asu Vanda Gene Industrial Research Company, Tehran 1533666398, Iran; Shaphagh@icloud.com; 4Resident of Orthopedics, Department of Orthopedics, School of Medicine, 5th Azar Hospital, Golestan University of Medical Sciences, Gorgan 4934174515, Iran; ah_zabolian@student.iautmu.ac.ir; 5Young Researchers and Elite Club, Tehran Medical Sciences, Islamic Azad University, Tehran 1916893813, Iran; h.saleki@student.iautmu.ac.ir (H.S.); Khoshbakht.ma1378@gmail.com (M.A.K.); Sinasabet771124@gmail.com (S.S.); M.j.naghdi1378@gmail.com (M.J.N.); 6Department of Food Hygiene and Quality Control, Division of Epidemiology, Faculty of Veterinary Medicine, University of Tehran, Tehran 1419963114, Iran; Kiavash.hushmandi@gmail.com; 7Faculty of Engineering and Natural Sciences, Sabanci University, Orta Mahalle, Üniversite Caddesi No. 27, Orhanlı, Istanbul 34956, Turkey; milad.ashrafizadeh@sabanciuniv.edu; 8Sabanci University Nanotechnology Research and Application Center (SUNUM), Istanbul 34956, Turkey; alizarrabi@sabanciuniv.edu; 9Department of Biology, Faculty of Science, Islamic Azad University, Science and Research Branch, Tehran 1477893855, Iran; Sepideh.mirzaei@srbiau.ac.ir; 10Department of Biomedical Engineering, Faculty of Engineering and Natural Sciences, Istinye University, Sariyer, Istanbul 34396, Turkey; 11Department of Pharmacology, Yong Loo Lin School of Medicine, National University of Singapore, Singapore 117600, Singapore; 12Cancer Translational Research Programme, Yong Loo Lin School of Medicine, National University of Singapore, Singapore 117600, Singapore

**Keywords:** medicinal herbs, cancer treatment, cancer stem cells, drug resistance, metastasis, proliferation

## Abstract

As a multifactorial disease, treatment of cancer depends on understanding unique mechanisms involved in its progression. The cancer stem cells (CSCs) are responsible for tumor stemness and by enhancing colony formation, proliferation as well as metastasis, and these cells can also mediate resistance to therapy. Furthermore, the presence of CSCs leads to cancer recurrence and therefore their complete eradication can have immense therapeutic benefits. The present review focuses on targeting CSCs by natural products in cancer therapy. The growth and colony formation capacities of CSCs have been reported can be attenuated by the dietary agents. These compounds can induce apoptosis in CSCs and reduce tumor migration and invasion via EMT inhibition. A variety of molecular pathways including STAT3, Wnt/β-catenin, Sonic Hedgehog, Gli1 and NF-κB undergo down-regulation by dietary agents in suppressing CSC features. Upon exposure to natural agents, a significant decrease occurs in levels of CSC markers including CD44, CD133, ALDH1, Oct4 and Nanog to impair cancer stemness. Furthermore, CSC suppression by dietary agents can enhance sensitivity of tumors to chemotherapy and radiotherapy. In addition to in vitro studies, as well as experiments on the different preclinical models have shown capacity of natural products in suppressing cancer stemness. Furthermore, use of nanostructures for improving therapeutic impact of dietary agents is recommended to rapidly translate preclinical findings for clinical use.

## 1. Introduction

The cancer is the second leading cause of death worldwide after cardiovascular diseases [1]. Based on the new estimates published by Siegel and co-authors, prostate cancer is the most common cancer in males, while breast cancer is the most common cancer in females. Noteworthy, lung cancer is the most aggressive cancer in both sexes and causes highest death among other tumors [2]. Regardless of cancer incidence rate and cell deaths, there have been efforts in developing novel therapeutics for tumor treatment [3,4,5,6]. Surgery or tumor resection is beneficial in early stages of cancer and when tumor cells diffuse into various tissues in body, it is impossible to eliminate cancer by using surgery [7,8]. Radiotherapy is less invasive compared to surgery, but it has its own problems including side effects and risk of resistance [9]. Immunotherapy is a new emerging therapeutic modality in cancer and uses checkpoint inhibitors in impairing cancer progression [10,11,12]. Another strategy is tumor treatment is chemotherapy that is most common compared to other modalities, but resistance and adverse impacts reduce its potential [13,14]. In order to prevent therapy resistance, combination cancer therapy has been utilized. In this strategy, a combination of chemotherapy and radiotherapy, or chemotherapy and immunotherapy are applied to suppress cancer progression and inhibit resistance [15,16]. Other kinds of treatments including photothermal therapy induced by nanoparticles in ablating tumor progression, inducing DNA damage and preventing cell cycle progression [17,18,19]. Besides, gene therapy using small interfering RNA (siRNA), short hairpin RNA (shRNA) and CRISPR/Cas9 system can be applied in cancer suppression [14,20,21,22]. These strategies have been partially advantageous in improving overall survival and prognosis of cancer patients. However, cancer is still a challenge for healthcare providers and new attempts should be made in this case [23,24,25,26,27,28,29,30].

The plant derived-natural products have been under attention in recent years in field of cancer therapy [31,32,33]. These agents have great therapeutic activities that anti-tumor activity is among them [34,35]. Due to the potential of phytochemicals in apoptosis induction, cell cycle arrest, metastasis inhibition and multitargeting capacity, they are able to inhibit cancer progression [20,36,37,38,39,40,41,42]. Clinical trials have shown that plant derived-natural products are generally well-tolerated in cancer patients [43,44]. Therefore, they can be extracted for developing commercialized drugs in cancer therapy [45]. Experiments have shown that natural occurring compounds are beneficial in sensitizing tumor cells to therapeutic modalities [46,47]. For instance, curcumin can increase sensitivity of cancer cells to cisplatin, docetaxel, paclitaxel and doxorubicin [31,48,49]. A combination of resveratrol and radiotherapy is advantageous in suppressing cancer progression [50]. Hence, application of natural products alone or in combination with other therapeutic strategies can pave the way towards effective tumor treatment [51,52,53]. The present review article focuses on using dietary agents in cancer therapy via targeting cancer stem cells (CSCs). For this purpose, we first provide a summary of CSCs, their analysis, markers and metabolism. Then, we show how CSCs can enhance stemness and progression of tumor cells. Finally, we mechanistically discuss how each phytochemical can be beneficial in cancer suppression via targeting CSCs.

## 2. Cancer Stem Cells

The stem cells have self-renewal capacity and can develop colonies [54,55]. The stem cells exist in various phases of life from embryonic phase to adulthood and are able to differentiation in forming various organs and tissues of body [56,57]. A kind of cells with characteristics similar to stem cells was found to be involved in carcinogenesis and called CSCs [58,59,60]. The CSCs are abundantly found in the tumor microenvironment and due to their self-renewal capacity, they can preserve population of tumor cells and mediating tumorigenesis. Besides, CSCs can differentiate into different cell kinds, enhancing cancer progression [61,62]. Overall, there are two concepts for carcinogenesis. At the first model, known as stochastic model, tumor cells are similar and have the same potential in tumorigenesis. Based on this model, the accumulation of mutations has resulted in carcinogenesis [61,63]. However, upon discovery of CSCs, a new concept of tumorigenesis was introduced, known as hierarchical model that CSCs are responsible for cancer development, maintaining and tumor seeding [64].

The identification of CSCs occurred in nineteenth century, when it was found that there are dormant cells in adult tissues that can be activated by stimuli and have capacity of proliferation and generating large masses of cells [65]. Although this was a great idea showing a special function for stem cells in cancer progression, it was ignored until in 1994 that Lapidot and colleagues isolated CSCs from leukemia cells and confirmed their presence [58]. The isolated CSCs were injected in mice and they showed potential in tumor initiation and development. After the isolation of CSCs from breast cancer in 2003 [60], more investigation was performed to isolated CSCs from other kinds of tumors including brain tumors, colorectal cancer and liver cancer [66,67,68].

Overall, CSCs have three distinct features from normal cells including differentiation, self-renewal capacity and homeostasis control [69,70]. Regardless of tumor stage, CSCs can be abundantly found in TME and a variety of techniques for isolation and enrichment of CSCs are utilized such as side population detection of cells with ability for Hoechst 33,342 exclusion, sphere forming capacity and aldehyde dehydrogenase (ALDH) measurement [71]. The Oct4, SOX2, Nanog, c-Myc and KLF4 are able to regulate CSC features and related signaling networks are Wnt, Notch, Hedgehog and PI3K/Akt, among others. Furthermore, complicated conditions in TME such as hypoxia, stromal cells, growth factors and extracellular matrix are able to control CSC characteristics in tumor cells [72,73,74,75,76,77].

One of the important aspects of CSCs is their metabolism. Based on experiments, CSCs rely on glycolysis, mitochondrial oxidative phosphorylation and other metabolic pathways that can be targeted therapeutically for suppressing CSCs [78,79,80]. It has been reported that CSCs are able to induce glycolysis via upregulating glucose transporters (GLUTs), hexokinases (HKs), monocarboxylate transporters and pyruvate dehydrogenase kinase 1 [81,82,83,84,85]. The CD133+ cells that exist in pancreatic cancer and glioma, have the ability of oxidative phosphorylation and enhancing expression level of genes involved in tricarboxylic acid (TCA) cycle [86,87]. Noteworthy, the levels of reactive oxygen species (ROS) can also affect CSC metabolism. The CSCs that are in dormant conditions can preserve low levels of ROS via glycolysis or stimulating antioxidant defense system. However, CSCs want to proliferate and differentiate, they induce ROS overgeneration via oxidative phosphorylation [88]. Therefore, metabolism, growth and differentiation of CSCs have a close association that should be considered.

## 3. Cancer Stem Cells in Oncology

The presence of CSCs is in favor of tumor progression. Thanks to experiments performed recently to shed some light on the role of CSCs. It seems that presence of CSCs in TME results in immune evasion and immunosuppression [89]. The CSCs are able to induce drug resistance feature of tumor cells. It has been reported that STAT3 signaling enhances stemness and CSC features in ovarian tumor cells to mediate their resistance to cisplatin and paclitaxel chemotherapy [90]. An interesting study has provided new insight about process of CSC generation in breast cancer and enhancing carcinogenesis. In this case, adipose-derived stem cells and breast cancer cells (MDA-MB-231 cells) fuse to produce CSCs. This process is mediated by CD44 [91]. As it was mentioned, CSCs involve in triggering drug resistance in tumor cells. Noteworthy, CSCs can also mediate radioresistance features. An experiment on nasopharyngeal cancer demonstrated that hTERT promotes CSCs features in nasopharyngeal cancer and mediates radioresistance. Knock-down of hTERT is correlated with reduced CSC characteristics and enhanced sensitivity to radiotherapy [92].

A variety of molecular pathways can modulate CSC features in tumors. For instance, microRNA (miRNA)-326 is suggested to be a tumor-suppressor in cervical cancer. The miRNA-326 binds to 3′-UTR of transcription factor 4 (TCF4) to diminish its expression. Upon TCF4 down-regulation, a significant decrease occurs in CSC features via down-regulating CD44 and SOX4 expression levels [93]. On the other hand, tumor-promoting factors pave the way for increasing CSC features in tumors. For instance, DUSP9 undergoes overexpression in triple-negative breast cancer and down-regulates ERK1/2 expression to enhance levels of CSC markers including SOX2, Oct4 and ALDH1 [94]. Therefore, recapitulation of CSC niche can promote tumor progression and mediate drug resistance feature [95]. Identification of such factors and reducing their expression can pave the way to drug sensitivity. For instance, silencing RAD51AP1 is associated with impairment in self-renewal capacity of CSCs and inducing drug sensitivity in colorectal cancer [96]. Another example is musashi-1 that enhances glioblastoma progression. Silencing musashi-1 diminishes CSC features in glioblastoma [97].

Each experiment has focused on a certain molecular pathway that leads to cancer progression via enhancing CSC features. The lung cancer stemness and CSC features can be mediated via JAK2/STAT3 axis. As upstream mediator, aryl hydrocarbon receptor induces JAK2/STAT3 axis to promote lung cancer stemness [98]. Another experiment reveals that long non-coding RNA (lncRNA)-WDFY3-AS2 reduces miRNA-139-5p expression to promote SCD4 expression. Then, ovarian cancer stemness enhances via promoting CSC features and resistance to cisplatin is mediated [99]. The NEDD4 expression undergoes upregulation in breast cancer and preserves stemness via promoting CSC features [100].

One of the important aspects of cancer progression is the role of extracellular vesicles (EVs), especially exosomes. Briefly, exosomes can provide cell–cell communication via transferring proteins, lipids and nucleic acids [101,102,103]. A recent study has shown that EVs can stimulate generation of CSCs from stem or progenitor cells [104]. The exosomes containing lncRNA UCA1 enhances SOX2 expression via miRNA-122-5p down-regulation to enhance differentiation and self-renewal capacity of CSCs [105]. It is worth mentioning that CSCs can also secrete exosomes in cancer progression. For instance, exosomes derived from CSCs contain lncRNA DOCK9-AS2 that can enhance growth, metastasis and stemness of thyroid tumor via inducing Wnt/β-catenin axis [106]. Overall, experiments highlight the fact that CSCs play a significant role in progression of tumors and their targeting is of importance in cancer therapy. Furthermore, a variety of molecular pathways including PCGF1, circFAM73A, CXCL1 and NUMB are able to affect CSC features. The growth, metastasis and therapy response are mainly regulated by CSC characteristics [107,108,109,110,111,112,113,114,115].

The interesting point is the role of CSC markers as prognostic factors in tumor [116]. The clinical studies have confirmed this statement. The overexpression of BMI-1 and CD44 as CSC markers occurs in head and neck squamous carcinoma to promote cancer progression and mediate undesirable prognosis [117]. The CD133 and CXCR4 as other CSC markers also demonstrate alterations in osteosarcoma patients. The CD133 upregulation occurs in 26% of patients, while CXCR4 demonstrates overexpression in 36% of cases. The overexpression of aforementioned CSC markers provides undesirable prognosis and survival of osteosarcoma patients [118]. 

## 4. Search Strategy

Various databases including Google scholar, Web of Science and Pubmed were used to search and collect articles. The names of dietary agents discussed in this article and other words including “cancer” and “cancer stem cell” were searched to find the relevant articles. Furthermore, there were many phytochemicals found whose impact on CSCs have not been evaluated yet but can be considered in future studies. 

## 5. Dietary Agents and Cancer Stem Cells

### 5.1. Flavonoids

#### 5.1.1. Flavones

##### Nobiletin

The nobiletin is a potent anti-tumor agent capable of suppressing tumor migration via EMT inhibition [119]. Nobiletin stimulates apoptosis and DNA damage to impair progression of oral cancer cells [120]. Nobiletin suppresses breast cancer progression in a dose-dependent manner. Nobiletin enhances miRNA-200b expression to elevate apoptosis and pyroptosis in breast cancer cells [121]. Three experiments have shown role of nobiletin in affecting CSCs in tumor therapy. The Wnt/β-catenin signaling is a possible target of nobiletin in impairing CSC characteristics [122]. The invasion and angiogenesis are suppressed by nobiletin via targeting CSCs. Nobiletin (100 and 200 μM) reduces STAT3 expression via binding to CD36 to inhibit NF-κB signaling, leading to a significant decrease in migration and metastasis of CSCs [123]. In order to potentiate efficacy of nobiletin in CSC suppression, its co-administration with xanthohumol is recommended. This combination suppresses migration of CSCs and decreases CD44v6 expression. Furthermore, they induce apoptosis and cycle arrest at G2/M phase. This combination impairs progression of colorectal CSCs and enhances their sensitivity to oxaliplatin and 5-flouroruacil chemotherapy [124].

##### Chrysin

The chrysin is a new emerging anti-tumor agent capable of suppressing growth and invasion of tumor cells, and promoting their sensitivity to chemotherapy [125]. A combination of chrysin and daidzein inhibits colorectal cancer progression via suppressing ERK and Akt molecular pathways [126]. Chrysin-loaded nanostructures are capable of enhancing apoptosis via triggering p53 expression. Furthermore, chrysin-loaded nanoparticles suppress PI3K/JNK axis and inhibit tumor growth in vivo [127]. Chrysin enhances miRNA-let-7a expression, while it reduces H19 and COPB2 expression levels to impair progression of gastric cancer cells [128].

An experiment has applied micellar nanoparticles for co-delivery of chrysin and docetaxel in cancer therapy. The application of micelles promotes therapeutic effect of both docetaxel and chrysin. The micelles were biodegradable and capable of docetaxel and chrysin co-delivery in synergistic cancer chemotherapy. This combination enhanced ROS levels to induce apoptosis in colon CSCs. Furthermore, docetaxel- and chrysin-loaded micelles inhibit migration and invasion of CSCs, impairing colon cancer metastasis [129]. Another study has focused on a derivative of chrysin, known as CHM-04 that has 3.2-fold higher anti-tumor activity compared to chrysin. The CHM-04 suppresses colony formation capacity and invasion of breast CSCs and induces apoptotic cell death [130].

##### Apigenin

Similar to chrysin, apigenin is a potent anti-tumor agent against various cancers including breast cancer, lung cancer and gastric cancer [7]. Apigenin impairs progression of multiple myeloma cells in a dose-dependent manner. Apigenin is able to down-regulate STAT1 expression in suppressing COX-2/iNOS axis [131]. A combination of apigenin and hesperidin prevent DNA repair in breast tumor to potentiate DOX activity in cancer suppression [132]. Furthermore, apigenin reduces activity of ABCG2 and ABCC4 as drug efflux transporters to enhance internalization of doxorubicin in breast cancer and mediate apoptosis [133].

The activation of YAP/TAZ axis is responsible for CSC features in triple negative breast cancer. The apigenin administration (0–64 μM) decreases colony formation and self-renewal capacity of CSCs. Furthermore, apigenin reduces number of CD44+ cells in breast cancer. These anti-tumor activities were mediated via suppressing YAP/TAZ axis [134]. The presence of CD133 cells decreases potential of cisplatin in lung cancer suppression. The apigenin administration (10–30 μM) induces apoptosis in CSCs via p53 upregulation and enhances cisplatin cytotoxicity against lung tumors [135]. The stimulation of tumor-promoting factors enhances CSC features in tumors. For instance, activation of PI3K/Akt signaling leads to CSC features in prostate cancer via inducing NF-κB signaling. The apigenin administration (0–100 μM) suppresses PI3K/Akt/NF-κB axis to reduce Oct3/4 levels, as CSC markers in prostate cancer. Apigenin reduces survival of CSCs and enhances p21 and p27 upregulation. For decreasing viability of CSCs in prostate cancer therapy, apigenin induces both intrinsic and extrinsic apoptosis. Furthermore, apigenin disrupts invasion and metastasis of CSCs [136]. When prostate cancer cells obtain stemness and CSC features, they can easily achieve resistance to cisplatin chemotherapy. Similar to previous experiment, apigenin (15 μM) suppressed phosphorylation of PI3K, Akt and NF-κB in impairing CSC features. Furthermore, apigenin inhibited cell cycle progression of CSCs in prostate cancer via upregulating p21, CDK2, CDK4 and CDK6. By reducing Snail expression, apigenin impaired progression and invasion of CSCs. Apigenin enhanced capase-8, Apaf-1 and p53 levels, while it decreased Bcl-2, sharpin and survivin levels in triggering apoptosis in CSCs in prostate cancer. These impacts of apigenin promote sensitivity of prostate cancer cells to cisplatin chemotherapy [137]. For disrupting cancer stemness, apigenin (40 μM) decreases expression levels of CD44, CD105, Nanog, Oct4, VEGF and REX-1 [138]. These studies demonstrate that apigenin is a potent inhibitor of CSCs in tumor treatment that further experiments can focus on revealing more signaling pathways affected by apigenin [139]. 

##### Baicalein

The baicalein is another anti-tumor agent that can induce apoptosis in tumors via upregulating caspase-3, -8 and -9 levels. Baicalein reduces MMP-2 and MMP-9 expression levels to impair metastasis of cancer cells [140]. The baicalein prevents SNO-induced ezrin tension and reduces iNOS levels to impair progression of lung tumors [141]. By suppressing Akt and Nrf2 molecular pathways, baicalein induce both apoptosis and autophagy in gastric tumor cells [142]. Furthermore, baicalein mediates proteasomal degradation of MAP4K3 to impair progression of lung tumor cells [143].

The members of Sonic Hedgehog signaling including SHH, SMO and Gli2 undergo upregulation to mediate CSC features in pancreatic cancer. The overexpression of Sonic leads to upregulation of SOX2 and Oct4 as CSC markers in pancreatic cancer. The baicalein administration (0–300 μM) suppresses Sonic signaling to reduce SOX2 expression and impair CSC features in pancreatic cancer [144].

##### Wogonin

The wogonin has demonstrated great therapeutic impacts in pre-clinical experiments [145]. The wogonin is able to induce senescent in breast tumors via down-regulating TXNRD2 expression [146]. Wogonin diminishes expression levels of Notch1 at mRNA and protein levels to suppress growth and metastasis of skin cancer cells [147]. This section focuses on wogonin impact on CSCs.

As it was mentioned, natural products can enhance ROS levels to induce apoptosis in CSCs [148]. A same strategy is followed by wogonin in osteosarcoma therapy. For this purpose, wogonin (0–80 μM) enhances ROS levels to reduce expression of factors responsible for CSC features such as STAT3, Akt and Notch1 [149]. In addition to triggering apoptosis and reducing survival of CSCs in osteosarcoma, wogonin is able to affect invasion of CSCs. In this way, wogonin administration (0–80 μM) decreases MMP-9 expression to impair migration and invasion of CSCs in osteosarcoma (Figure 1) [150]. Interestingly, studies have only focused on osteosarcoma and more experiments on other tumor models should be performed to shed more light on anti-tumor activities of wogonin via targeting CSCs. Overall, flavones are potential agents in suppressing stemness and CSC features in tumors that have been summarized in Table 1.

#### 5.1.2. Flavanones

**Naringenin** administration (100 μM) prevents colony formation, metastasis and EMT in breast tumor, while it induces apoptosis via upregulating p53 and ERα at mRNA level, as tumor-suppressor factors [152].

**Naringin** administration (300 μM) impairs CSC features in esophageal cancer and inhibit viability of CSCs [153].

**Hesperetin** (50–200 μM) also demonstrated potential in triggering apoptosis in breast CSCs. Hesperetin impairs invasion of breast CSCs and stimulates cell cycle arrest in breast CSCs. The hesperetin enhances p53 expression, while it down-regulates Notch1 expression in impairing stemness in breast CSCs [154].

The experiments evaluating role of flavanones in targeting CSCs in tumor suppression are limited and more studies are required to reveal true potential of these natural products in cancer therapy. 

#### 5.1.3. Flavonols

##### Fisetin

The fisetin is a natural flavonol that has demonstrated high anti-tumor activity and capacity in chemoprevention [155,156]. Fisetin mediates histone demethylation to induce DNA damage and impair progression of pancreatic tumor cells [157]. To date, just one experiment has evaluated role of fisetin in targeting CSCs that is included here. The proliferation, metastasis, angiogenesis and carcinogenesis of renal CSCs undergo inhibition by fisetin. The in vitro and in vivo experiments have shown role of fisetin in suppressing renal CSCs. The fisetin is able to decrease expression levels of cyclin Y and CDK16 via inhibiting 5hmC modification in CpG islands. Furthermore, fisetin can reduce TET1 levels in renal CSCs. Therefore, a significant decrease occurs in growth of renal CSCs and their angiogenesis and migration are suppressed [158].

##### Epigallocatechin 3-Gallate

The epigallocatechin 3-gallate (EGCG) is another naturally occurring compound that has demonstrated high potential in cancer therapy. The EGCG enhances Beclin and LC3 levels to induce autophagy in bladder cancer cells. Furthermore, EGCG induces apoptosis in bladder cancer via upregulating Bax, caspase-3 and caspase-9 levels [159]. The EGCG reduces Sonic Hedgehog expression to suppress PI3K/Akt axis, leading to apoptosis in colon cancer cells [160]. The EGCG impairs migration and metastasis of cervical tumor cells via down-regulating TGF-β and subsequent inhibition of EMT mechanism [161]. Therefore, EGCG can be considered as a promising agent in cancer therapy [162]. Noteworthy, EGCG targets CSCs in affecting cancer progression. The lung tumor cells demonstrate high expression level of CLOCK to improve their CSC features. Noteworthy, EGCG (0–40 μM) reduces mRNA and protein levels of CLOCK to impair CSCs features and suppress self-renewal capacity of CSCs in lung tumor therapy [163]. The miRNAs are considered as important modulators of CSCs in cancer [164,165,166,167]. The expression level of hsa-miRNA-485-5p undergoes down-regulation in serum of lung tumor patients, showing tumor-suppressor role of this miRNA. Restoring miRNA-485-5p expression impairs lung tumor proliferation and stimulates apoptosis via down-regulating RXRα expression. The EGCG administration (0–40 μM) enhances miRNA-485-5p expression to reduce RXRα expression, leading to a decrease in expression levels of CD133 and CD44 as CSC markers [168]. The CSC features in lung cancer mainly depend on upregulation of β-catenin and its inhibition suppresses lung tumor progression. An experiment has shown that EGCG (0–100 μM) inhibits Wnt/β-catenin axis to impair CSC features in lung cancer [169].

Similar to lung cancer, a number of experiments have focused on anti-tumor activity of EGCG in colorectal cancer via targeting CSCs. In this way, EGCG (0–40 μM) suppresses Wnt/β-catenin axis to reduce expression levels of CD133, CD44, ALDHA1, Nanog and Oct4 in impairing CSC features in colorectal tumor [170]. The colorectal tumor cells demonstrate resistance to various chemotherapeutic agents including oxaliplatin and 5-flourouracil [171,172]. The EGCG (0–400 μM) decreases expression levels of Notch1, Bmi1, Suz12 and EZH2, while it enhances expression levels of miRNA-34a, miRNA-145 and miRNA-200c in impairing CSC features in colorectal cancer and enhancing 5-flourouracil sensitivity [173]. The bladder cancer cells rely on Sonic Hedgehog signaling to enhance their CSC features and mediate their progression. Noteworthy, EGCG suppresses Sonic signaling to decrease CD44, CD133, Nanog, Oct4 and ALDH1 in bladder cancer therapy [174]. Therefore, EGCG is a potent inhibitor of CSCs in various cancers and for this purpose, it targets various molecular pathways including Notch and NF-κB signaling pathways (Table 2) [175,176,177].

#### 5.1.4. Chalcones

##### Isoliquiritigenin

The isoliquiritigenin (ISL) is derived from licorice root and it is a modulator of molecular pathways in cancer suppression [178]. The ISL suppresses TGF-β signaling to reduce Smad3 expression, resulting in EMT inhibition and decreased endometrial cancer migration [179]. Furthermore, ISL enhances miRNA-200c expression to inhibit β-catenin signaling, leading to EMT suppression in triple-negative breast cancer [180]. A combination of ISL and docosahexaenoic acid induces apoptosis in colorectal cancer via enhancing ROS levels and mediating phosphorylation of JNK and ERK [181]. The ISL-loaded liposomes suppress colorectal tumor progression via inhibiting AMPK-mediated glycolysis [182]. This section focuses on anti-tumor activity of ISL via targeting CSCs. The ISL administration (25 μM) suppresses self-renewal and multidifferential capacities of CSCs in breast cancer. Furthermore, ISL inhibit growth and colony formation of breast CSCs. The ISL suppresses β-catenin signaling and decreases ABCG2 expression to enhance drug sensitivity of breast cancer cells and suppress CSC features [183]. The expression levels of CD44 and ALDH1 undergo down-regulation upon ISL administration (0–50 μM) in oral cancer. The ISL suppresses colony formation and invasion of CSCs in oral tumor. In suppressing CSCs, ISL also decreases expression level of GRP78 at mRNA and protein levels. Furthermore, by reducing ABCG2 expression, ISL enhances sensitivity of oral CSCs to cisplatin chemotherapy [184]. Another experiment reveals that ISL administration (25 and 50 μM) enhances SIF1 expression via demethylation to suppress DNMT1 methyltransferase. Furthermore, ISL simultaneously suppresses β-catenin signaling to eradicate CSCs in breast cancer therapy [185]. 

#### 5.1.5. Isoflavonoids

##### Daidzein

An experiment has focused on a derivative of daidzein, known as N-t-boc-Daidzein. This derivative is able to suppress CSC features in ovarian cancer in concentration (0–10 μM)- and time (0–70 h)-dependent manner. This derivative induces apoptosis in CSCs via upregulating caspase-3, -8 and -9 levels. Suppressing growth and survival of CSCs is mediated by N-t-boc-Daidzein. In addition to triggering caspase cascade, N-t-boc-Daidzein mediates mitochondrial depolarization and decreases Akt expression via degradation to impair CSC features in ovarian tumor [186].

##### Genistein

The genistein is an isoflavonoid compound that has potent anti-tumor activity. The genistein administration decreases chance of mammary cancer recurrence [187]. The glioblastoma cells exposed to radiation, demonstrate enhanced metastasis. The genistein is able to suppress Akt2 pathway in reducing invasion and migration of glioblastoma cells [188]. The genistein administration reduces hsa-circ-0031250 expression to promote miRNA-873-5p expression, leading to lung cancer suppression and decreased proliferation rate [189]. Both inflammation and GSK-3 are suppressed by genistein to inhibit ovarian tumor progression in vivo [190]. In order to improve genistein capacity in cancer suppression, nanoparticles have been developed for its targeted delivery [191,192].

The Sonic Hedgehog signaling enhances CSC features in nasopharyngeal cancer. The administration of genistein (0–100 μM) inhibits sonic signaling and reduces expression level of CSC markers including CD44, ALDH1, Oct4 and Nanog to impair nasopharyngeal progression [193]. A same strategy occurs in renal cancer, so that genistein administration (0–90 μM) suppresses Sonic Hedgehog signaling to impair growth and colony formation of renal CSCs and stimulate apoptotic cell death [194]. The genistein is able to mediate differentiation of CSCs in a paracrine mechanism to suppress cancer progression. An experiment on breast tumor has shown genistein administration (2 μM or 40 nM) stimulates PI3K/Akt and MEK/ERK molecular pathways to induce differentiation of CSCs in breast cancer [195]. Another experiment also reveals that low doses of genistein (15 μM) inhibits self-renewal capacity of gastric CSCs in vitro and in vivo [196]. The capacity of genistein in suppressing CSC features in gastric cancer is attributed to down-regulating FOXM1 expression that subsequently, reduces expression levels of CD133, CD44 and Nanog. Furthermore, by inhibiting FOXM1 expression, genistein reduces Twist1 expression that is in favor of inhibiting EMT in CSCs via reducing N-cadherin levels and enhancing E-cadherin levels [197]. In addition to aforementioned molecular pathways, genistein suppresses Gli1 expression to impair CSC features in gastric cancer [198]. Some of the molecular pathways are similar among various cancers. For instance, previous research demonstrated that genistein suppresses Gli1 pathway in reversing CSC features in gastric cancer. Another experiment in prostate cancer reveals that genistein is able to suppress Gli1 pathway in inhibiting CSC characteristics [199]. Furthermore, it was mentioned that genistein suppresses nasopharyngeal CSCs via inhibiting Sonic signaling. A similar study reveals Hedgehog signaling inhibition and subsequent Gli1 down-regulation by genistein in prostate cancer stemness suppression [199]. Genistein also suppresses Hedgehog signaling in impairing CSC features in breast cancer [200]. Overall, studies highlight the fact that genistein is a potent inhibitor of cancer progression via targeting CSCs (Figure 2, Table 3) [200,201,202,203,204].

### 5.2. Pomegranate and Its Bioactive Compounds

#### 5.2.1. Ellagic Acid

Ellagic acid is a polyphenolic compound that can suppress cancer progression. The ellagic acid increases sensitivity of bladder cancer cells via inhibiting drug transporters and EMT mechanism [206]. The p21 and p53 levels undergo upregulation by ellagic acid to induce apoptosis in prostate cancer cells [207]. The ellagic acid stimulates autophagy via Akt down-regulation and AMPK upregulation to reduce viability and survival of ovarian cancer cells [208]. The ellagic acid is a potent agent in suppressing drug resistance via down-regulating P-glycoprotein (P-gp_levels [209]. This section focuses on anti-tumor activity of ellagic acid based on targeting CSCs.

Ellagic acid is able to target CSCs in tumor eradication. An experiment has shown role of ellagic acid in breast cancer therapy via targeting CSCs. The β-catenin stabilization is responsible for CSC features in breast cancer and ACTN4 functions as upstream mediator. Silencing ACTN4 impairs growth, migration and colony formation capacities in breast cancer. The ellagic acid administration (0–50 μM) reduces ACTN4 expression to inhibit β-catenin signaling in CSCs, impairing breast cancer progression [210]. The ALDH level as an inducer of drug resistance and CSC marker undergoes down-regulation by a combination of ellagic acid and urolithins in colon CSCs [211]. Although just two experiments have evaluated role of ellagic acid in targeting CSCs, these obviously demonstrate role of ellagic acid in tumor treatment and more studies are required in this case. 

#### 5.2.2. Caffeic Acid

Caffeic acid (CA) is a phenolic acid compound that is isolated from natural sources such as tea and has various therapeutic activities. The anti-tumor activity of CA and its derivatives have been under attention, summarized in this review [212]. The CA can mediate ROS overgeneration to induce cell death in cervical cancer cells [213]. Furthermore, CA suppresses P-gp function to mediate chemosensitivity of tumors [214]. Noteworthy, CA derivatives are also able to induce apoptosis via survivin down-regulation and caspase-3 and -9 upregulation [215,216].

The colorectal CSCs that have high ability in mediating radioresistance feature. The CD44+ and CD133+ CSCs in colorectal cancer have self-renewal capacity and carcinogenesis impact. The CA administration inhibits PI3K/Akt axis that is responsible for CSCs features in colorectal cancer, leading to enhanced radiosensitivity and reduced carcinogenesis impact [217]. The keratinocytes demonstrate high expression level of NF-κB that is beneficial for upregulating Snail expression and mediating migration and CSCs features. The CA administration (0–100 μM) enhances p38 expression to reduce potential of NF-κB in binding to Snail. Therefore, invasion and CSC features undergo inhibition [218]. The CA has a naturally occurring derivative, known as caffeic acid phenethyl ester (CAPE) that can suppress self-renewal capacity and progenitor formation in breast cancer in a dose-dependent manner [219]. The expression of miRNAs is mainly affected by CA in cancer therapy. The TGF-β/Smad2 axis mediate CSC features in tumors. The CA administration (20 μM) increases miRNA-184a expression via DNA methylation, as a tumor-suppressor factor. Then, overexpressed miRNA-184a inhibits Smad2 expression by binding to its 3′-UTR, impairing CSC features [220]. Hence, CA and its derivative CAPE are potential anti-tumor agents in cancer treatment via targeting CSCs.

#### 5.2.3. Luteolin

The luteolin is a natural flavonoid that can suppress cancer progression via regulating autophagy [221]. The luteolin suppresses Akt/mTOR axis to reduce MMP-9 expression, impairing metastasis and invasion of breast cancer cells [222]. The luteolin suppresses EMT mechanism via down-regulating YAP/TAZ, resulting in a significant decrease in breast cancer invasion [223]. Furthermore, lutelon enhances death receptor 5 (DR5) expression and mediates mitochondrial fission to promote TRAIL sensitivity of lung tumors [224]. To date, just one experiment has evaluated role of luteolin in targeting CSCs. Mechanistically, IL-6 stimulates STAT3 signaling to promote CSC features in oral cancer and mediate their resistance to radiotherapy. The luteolin administration (0–40 μM) suppresses IL-6/STAT3 axis to impair self-renewal capacity of CSCs and reduce expression level of CSC markers including ALDH1 and CD44 [225]. 

#### 5.2.4. Quercetin

Quercetin is a flavonol present in fruits and vegetables such as grape, anion, apple, berries and broccoli [226]. The diet intake of quercetin is estimated to be higher than 70% of all flavonol intake [227]. Quercetin has different pharmacological activities including immunomodulatory, hepatoprotective, neuroprotective and nephroprotective that can be mediated via regulating autophagy [228]. Furthermore, quercetin can be considered as a protective agent against ischemic/reperfusion injury [229]. Regardless of its protective impacts, quercetin is suggested to display significant anti-tumor activity. Quercetin reduces CDK6 expression to impair progression of breast and lung cancer cells [230]. Quercetin enhances reactive oxygen species (ROS) levels to induce ferroptosis. Furthermore, quercetin promotes cell death via lysosome activation [231]. Angiogenesis and metastasis of esophageal cancer cells was suppressed by quercetin via reducing expression levels of VEGF-A, MMP-2 and MMP-9 [159]. Quercetin reverses multidrug resistance and using nanoparticles for its delivery enhances its potential in cancer suppression [232]. This section focuses on CSC targeting by quercetin in cancer therapy.

The PI3K/Akt/mTOR axis is responsible for growth and progression of CSCs in breast cancer. The quercetin administration (0–200 μM) suppresses PI3K/Akt/mTOR axis in induce cell cycle arrest in G1 phase in breast cancer cells. The quercetin impairs viability, colony formation and mammosphere formation in CD44+ stem cells in breast cancer and triggers apoptosis [233]. Another experiment also reveals role of quercetin in eradicating CSCs in breast cancer. The quercetin administration (0–200 μM) suppresses growth, metastasis and self-renewal capacity of CSCs in breast cancer. For this purpose, quercetin reduces expression levels of ALDH1, CXCR4, mucin 1 and epithelial cell adhesion molecule (EpCAM) [234]. The nuclear translocation of Y-box binding protein 1 (YB-1) is responsible for CSCs features in breast cancer. For reversing multidrug resistance in breast cancer, quercetin decreases expression level of P-gp as an efflux transporter. More importantly, quercetin inhibits nuclear translocation of YB-1 to suppress CSCs features in breast cancer cells and enhance their sensitivity to doxorubicin, paclitaxel and vincristine [235]. It is worth mentioning that by down-regulating expression levels of P-gp, BCRP and MRP1, quercetin (0–2 μM) increases internalization of doxorubicin in breast cancer cells, resulting in eradication of CSCs [236].

Another cancer that can be affected by quercetin is pancreatic cancer. For enhancing potential of quercetin in suppressing CSCs, its combination therapy has been used. For this purpose, a combination of sulforaphane (SFN) (0–10 μM) and quercetin (20 μM) effectively inhibits self-renewal capacity of CSCs [237]. In pancreatic cancer, CSC divisions tend to be symmetric. The low expression of miRNA-200b-3p induces Notch signaling to make daughter cells be symmetric. However, quercetin administration (50 μM) enhances expression level of miRNA-200b-3p as tumor-suppressor to reduce Notch expression, resulting in asymmetric divisions in CSCs [238]. A combination of SFN, quercetin and catechins reduces ALDH1 expression in pancreatic cancer, as CSC marker. For this purpose, SFN, quercetin and catechins enhances miRNA-let-7 expression to down-regulate K-ras expression [239]. Hence, quercetin and its combination with anti-tumor agents synergistically suppress CSC features in pancreatic cancer [240].

The induction of Notch1 signaling and enhanced CSC features lead to radioresistance in colon cancer. The quercetin administration (20 μM) suppresses Notch1 signaling and reduces expression levels of CSC markers including SOX9, CD133 and CD44 [241]. The CD133+ colorectal cancer cells seem to be responsible for triggering drug resistance in colorectal cancer. The quercetin administration (10–100 μM) induces apoptosis and cycle arrest (G2/M phase) in CD133+ cells to enhance sensitivity of colorectal tumor cells to doxorubicin chemotherapy [242]. Therefore, quercetin targets CSCs to increase therapy response of colorectal cancer cells.

The prostate cancer (PCa) is the most common cancer and second malignant tumor in men [2,243]. The various molecular pathways are responsible for PCa progression and various anti-tumor compounds and genetic tools have been utilized for its treatment [20,244,245,246,247]. The quercetin has shown capacity in suppressing growth and invasion of PCa stem cells (CD44+/CD133+ cells). The midkin (MK) pathway is responsible for CSC features in PCa. Co-application of quercetin and siRNA-MK enhances potential in suppressing CSCs. Furthermore, this combination induces apoptosis and G1 arrest in CSCs. The molecular pathways including PI3K/Akt, ERK1/2, p38, ABCG2 and NF-κB undergo down-regulation by quercetin and siRNA-MK in inhibiting CSCs in PCa [248]. Furthermore, a combination of quercetin and EGCG induces apoptosis (via caspase-3/7 upregulation and down-regulating Bcl-2, survivin and XIAP) in CSCs and prevents their growth and invasion. The reduced metastasis of CSCs by quercetin and EGCG is attributed to suppressing epithelial-to-mesenchymal transition (EMT) via down-regulating Snail, vimentin, Slug and β-catenin [249]. Therefore, quercetin is a well-known compound in eradicating CSCs in preventing cancer progression (Figure 3, Table 4) [250,251]. 

### 5.3. Carotenoids

#### 5.3.1. Astaxanthin

The astaxanthin is derived from algae and has anti-tumor activity. Astaxanthin The astaxanthin impairs gastric cancer development and progression via suppressing inflammatory storm [252]. Astaxanthin induces apoptosis and promotes PARPγ expression to reduce esophageal cancer progression [253]. The pontin overexpression is responsible for CSC features in breast cancer. Suppressing pontin expression impairs colony formation and CSC characteristics in breast cancer. The astaxanthin administration (80 and 100 μM) reduces pontin expression and impairs CSC features via down-regulating Oct4, Nanog and mutp53 levels [254].

#### 5.3.2. β-Carotene

The β-carotene is another member of carotenoids that is able to effectively suppress cancer progression. The β-carotene suppresses IL-6/STAT3 axis and inhibits M2 polarization of macrophages to impair colon tumor progression [255]. β-carotene induces apoptosis and reduces antioxidant markers to impair breast tumor progression [256]. A combination of β-carotene and lycopene suppresses esophageal cancer progression via down-regulating COX-2 and cyclin D1 expression levels [257]. For promoting potential of β-carotene in breast cancer suppression, lipid–polymer nanoparticles have been developed for its targeted delivery [258]. Compared to astaxanthin, more experiments have focused on role of β-carotene in targeting CSCs. The β-carotene administration (20 and 40 μM) reduces growth and colony formation capacities of colon CSCs. The β-carotene reduces DNMT3A expression at mRNA levels and prevents DNA methylation. Furthermore, β-carotene promotes histone H3 and H4 acetylation levels in impairing CSC progression [259]. The self-renewal capacity of CSCs undergoes a decrease by β-carotene in neuroblastoma cells and this leads to enhanced sensitivity of tumor cells to cisplatin chemotherapy [260]. Furthermore, in vivo experiment on xenograft model has shown role of β-carotene in suppressing CSC features in neuroblastoma [261]. The Oct3/4 and DLK1 undergo down-regulation by β-carotene in decreasing CSC features and stemness of neuroblastoma cells [262]. The potential of β-carotene in reducing DLK1 expression and impairing CSC features is mediated via retinoic acid receptor β [263]. 

### 5.4. Sulforaphane

The sulforaphane (SFN) is an isothiocyanate and is present in various vegetables including cabbage broccoli, cauliflower and Brussels. The SFN is generated upon glucoraphanin hydrolysis. The SNF application in cancer therapy has witnessed a growing increase [264]. A recent experiment demonstrates that SFN reduces expression level of H19, as a lncRNA to suppress growth and invasion of pancreatic cancer cells [265]. The SFN is able to suppress tubulin polymerization in triggering apoptosis and cell cycle arrest in glioblastoma cells [266]. The expression level of FAT-1 undergoes down-regulation by SFN to impair proliferation and invasion of bladder cancer cells [267]. The SFN administration is of interest in increasing sensitivity of breast cancer cells to doxorubicin chemotherapy via inhibiting myeloid-derived suppressor cells [268]. Therefore, SFN is a potent anti-tumor agent [269] and this section focuses on CSC targeting by SFN in cancer treatment.

The activation of Sonic Hedgehog signaling promotes stemness of gastric cancer cells. The SFN administration (0–10 μM) stimulates apoptosis in CSCs and inhibits their growth to suppress gastric cancer proliferation. For this purpose, SFN inhibits Sonic signaling [270]. The upregulation of TAp63α occurs in colorectal tumor spheres with CSC features. Noteworthy, TAp63α enhances self-renewal capacity and CSC markers in colorectal tumor. Investigation of molecular pathways reveals that SFN increases Lgr5 expression by binding to its promoter, leading to β-catenin signaling activation. The SFN administration (0–10 μM) reduces TAp63α expression to inhibit expression of CSC markers such as CD133, CD44, Nanog and Oct4 [224].

Human tumor necrosis factor (TNF)-related apoptosis ligand (TRAIL) is suggested to be involved in triggering apoptosis in tumor cells [271]. However, experiments have shown capacity of tumor cells in obtaining resistance to TRAIL-mediated apoptosis [272,273,274,275]. Furthermore, TRAIL can surprisingly enhance progression of cancer cells via inducing NF-κB signaling [276,277]. An experiment has shown synergistic impact between TRAIL and SFN in PCa therapy. Both TRAIL and SFN are able to suppress CSCs in PCa, and SFN has higher ability compared to TRAIL. However, TRAIL induces NF-κB signaling and promotes PCa progression. The SFN administration (10 μM) decreases expression level of NF-kB, CXCR4, Jagged1, Notch-1, SOX2, Nanog and ALDH1 to impairs CSC features and stemness in PCa [278]. It is worth mentioning that down-regulation of ALDH1, c-Rel and Nothc-1 as CSC markers by SFN, significantly elevates sensitivity of prostate and pancreas cancer cells to cisplatin, gemcitabine and 5-flourouracil chemotherapy [279]. The stimulation of NF-κB signaling is responsible for ALDH1 upregulation and enhanced CSC features in pancreatic cancer. Therefore, NF-κB inhibition by SFN (20 μmol/L) impairs CSC features in pancreatic cancer cells and enhances their sensitivity to sorafenib chemotherapy [280]. Overall, the experiments advocate the fact that SFN is a potent agent in targeting CSC and suppressing stemness of cancers (Figure 4, Table 5) [281,282,283,284,285].

### 5.5. Curcumin

The *curcuma longa* is a medicinal plant belonging to family of Zingiberaceae family that has a well-known bioactive compound, curcumin that can be isolated from its rhizome [294,295]. The curcumin and other curcuminoids lead to yellow color of rhizome. Although content of curcuminoids in rhizome of *curcuma longa* is various, it seems that curcumin is the main component and comprises 77% [296]. The curcumin has demonstrated various therapeutic activities that among them, anti-cancer potential is of importance [297]. The curcumin administration is beneficial in suppressing proliferation and metastasis of cancer cells. A recent experiment has shown role of curcumin in down-regulating NF-κB, ERK, MMP-2 and MMP-9, while it can enhance p38 and JNK levels to induce cell death and prevent invasion of leukemia cells [298]. Recently, attention has been directed towards chemopreventive role of curcumin in cancer therapy [299]. The curcumin- and docetaxel-loaded micellar nanoparticles effectively suppress progression of ovarian cancer cells via suppressing angiogenesis and inducing apoptosis. The curcumin is able to sensitize ovarian cancer cells to docetaxel chemotherapy [300]. Due to poor bioavailability of curcumin, nanostructures are mainly applied for curcumin delivery alone or its combination with other anti-tumor agents to exert synergistic cancer therapy [31,301]. A fibrin matrix has been developed for prolonged release of curcumin and suppressing cancer metastasis. As studies have shown role of the nanoplatforms for promoting therapeutic activity of curcumin, future clinical trials can be performed [302]. The current section focuses on anti-tumor activity of curcumin based on targeting CSCs.

The capacity of curcumin in cancer suppression via targeting CSCs is attributed to affecting various molecular pathways. The activation of Wnt/β-catenin and Sonic Hedgehog molecular pathways results in CSC features in lung cancer and enhancing tumor progression. The curcumin administration (0–40 μM) induces apoptosis in CSCs and diminishes their proliferation. Furthermore, curcumin reduces CSC hallmarks such as Nanog, Oct4, CD133, CD44 and ALDH1A1. Mechanistically, these anti-tumor activities of curcumin are mediated via inhibiting Wnt and Hedgehog signaling pathways [303]. In addition, nuclear factor-kappaB (NF-κB) pathway also participates in enhancing survival of CSCs [304]. The down-regulation of NF-κB signaling impairs CSC features in bladder cancer [305]. In liver cancer cells, curcumin reduces their proliferation and colony formation capacities as well as inhibiting CSCs features. The curcumin capacity in suppressing CSCs features in liver cancer is pertained to inhibiting NF-κB signaling [306]. Hence, identification of molecular pathways affected by curcumin in suppressing CSC features can broaden our understanding towards underlying mechanisms of its anti-tumor activity.

The signal transducer and activator of transcription 3 (STAT3) is another tumor-promoting factor related to cancer progression [307,308,309]. The upregulation of STAT3 enhances growth and invasion of tumor cells and mediates their therapy resistance [310,311,312,313,314,315,316]. The activation of STAT3 signaling promotes CSC features in tumors and its inhibition, for instance by acetaminophen, reduces CSC markers and colony formation capacity [317,318]. The curcumin administration (0–40 µM) decreases levels of CD44, ALDH, SOX2, Nanog and c-Myc as CSC markers to impairs thyroid cancer progression and induce apoptosis. Furthermore, by reducing CSC features, curcumin enhances potential of cisplatin in thyroid cancer suppression [319]. Similar phenomenon occurs in bladder cancer, so that curcumin administration (0–50 µM) decreases CD44, CD133, ALDH1, Nanog and Oct4 to impair CSC characteristics in bladder cancer and induce apoptosis and proliferation inhibition. The investigation of molecular pathways reveals that these anti-tumor activities are mediated via sonic signaling inhibition [320]. The interesting point is the curcumin and CD44 coupling in cancer therapy. An experiment demonstrates that colon cancer cells overexpressing CD44 as CSC marker are more sensitive to curcumin compared to CD44- cells. The curcumin-CD44 coupling stimulates apoptosis in colorectal stem cells via preventing influx of glutamine into cancer cells and reducing its intracellular accumulation [321].

In order to potentiate anti-tumor activity of curcumin, its combination with other tumor-suppressor agents is performed. A combination of curcumin and quinacrine can be beneficial in impairing breast cancer progression via targeting CSCs. The quinacrine is able to induce DNA damage in cancer cells, but its intracellular accumulation should be improved. In this case, curcumin is helpful and by suppressing ABCG2 activity via occupying its ligand-binding site, enhances internalization of quinacrine, resulting in cell death and DNA damage in breast CSCs [322]. Furthermore, curcumin also has capacity of inducing DNA damage in CSCs [323]. The curcumin can negatively affect colony formation capacity of lung cancer cells. It is suggested that curcumin prevents self-renewal capacity of CSCs to suppress colony formation of lung cancer cells [323]. Therefore, curcumin is a potent suppressor of CSCs in lung cancer and for this purpose, it can inhibit JAK2/STAT3 axis to impair CSC features [324]. Based on previously discussed molecular pathways, both STAT3 and NF-κB pathways are suppressed by curcumin in targeting CSCs. Noteworthy, STAT3 can function as upstream mediator of NF-κB and mediates its activation via enhancing IKKα stability [325]. A combination of curcumin (10 μM) and epigallocatechin gallate (10 μM) suppresses STAT3/NF-κB axis to impair CSC features and reduce number of CD44+ cells [326]. In addition to curcumin, its derivatives have been also capable of targeting CSCs. The colon cancer cells positive for ALDH and CD133 demonstrate activation of STAT3 signaling and its phosphorylation. An analogue of curcumin, known as GO-Y030 can suppress STAT3 phosphorylation to impair CSC features in colon cancer and prevent its progression [327]. Therefore, curcumin and its derivatives suppress proliferation and invasion of tumors via targeting CSCs. Furthermore, they can enhance sensitivity of cancer cells to therapies by targeting CSCs [328]. Interestingly, curcumin can be advantageous in enhancing sensitivity of cancer cells to radiotherapy [329]. The breast cancer stem-like cells have the ability of obtaining radioresistance. A combination of curcumin and glucose gold nanoparticles induces apoptosis in CSCs and reduces expression levels of hypoxia inducible factor-1α (HIF-1α) and HSP90 in sensitizing breast cancer cells to radiotherapy [330]. Therefore, curcumin is a versatile agent in suppressing CSC features in cancer and various molecular pathways such as Wnt, Sonic, STAT3 and NF-κB, among others, are affected to reduce CSC markers such as CD44, ALDH and CD133 (Figure 5, Table 6) [331,332,333,334,335,336,337,338]. 

### 5.6. Resveratrol

The resveratrol (Res) is a well-known compound in treatment of different ailments including diabetes, cancer and neurological disorders [340,341]. The recent years have witnessed an increase in attention towards Res application in cancer therapy. Res induces apoptosis via impairing mitochondrial function in cancer cells. Furthermore, tumor-promoting factors such as NF-κB, COX-2 and PI3K undergo down-regulation by Res in tumor suppression [342]. A combination of Res and radiation stimulates apoptosis in breast cancer via Bcl-2 down-regulation and Bax upregulation [50]. Res suppresses phosphorylation of STAT3 at tyrosine 705 to impair metastasis of cervical cancer cells [343]. Down-regulation of HPV E6 and E7 by Res leads to apoptosis and cell cycle arrest in cervical cancer [344]. Furthermore, Res inhibits progression and growth of PCa cells via suppressing PI3K/Akt signaling [345]. The c-Myc as downstream target of PI3K/Akt axis is suppressed by Res in lung cancer therapy [346]. Loading Res on nanostructures significantly promotes its potential in thyroid cancer suppression. Furthermore, Res-loaded nanoparticles decrease tumor volume up to 55% in thyroid cancer in vivo [347].

The tumor sphere formation capacity in renal cancer is mediated by CSCs and Sonic signaling plays a significant role in this case. The Res administration (0–30 μM) down-regulates Sonic signaling to inhibit CSCs features in renal cancer, leading to apoptosis and proliferation inhibition [348]. As it was mentioned earlier, Res is loaded on nanoparticles to promote its anti-tumor activity [349,350,351]. Another promising strategy for enhancing bioavailability of Res in synthesizing analogues with superior activities. The pterostilbene is a derivative of Res that has higher bioavailability compared to Res. The pterostilbene is able to reduce expression levels of CD133, Oct4, SOX2, Nanog and STAT3 in impairing CSC features and retarding progression of cervical cancer [352]. Furthermore, Res is able to suppress Wnt/β-catenin in reducing CSC features in breast cancer. It is worth mentioning that Res enhances LC-3II, Beclin-1 and ATG7 expression levels in triggering autophagy in CSCs and reducing breast cancer progression [353]. It is worth mentioning that autophagy has both tumor-promoting and tumor-suppressor roles in cancer [221,354,355] and its induction by Res requires more clarification.

The STAT3 signaling is considered as a tumor-promoting factor in osteosarcoma. The upregulation of STAT3 by VEGFR2 results in metastasis of osteosarcoma cells [356]. Furthermore, induction of JAK2/STAT3 axis by exosomal LCP1 leads to carcinogenesis and migration of osteosarcoma cells [357]. The Res administration (0–3 μM) decreases cytokine generation and suppresses JAK2/STAT3 axis to impair CSCs features in osteosarcoma via CD133 down-regulation [358]. Another molecular pathway that is responsible for CSCs features is nutrient-deprivation autophagy factor-1 (NAF-1). The Res inhibits NAF-1 signaling to reduce CSC features including SOX2, Oct4 and Nanog in pancreatic cancer treatment [359]. The interesting point is the interaction between tumor microenvironment (TME) components and molecular pathways in providing CSCs features. The T lymphocytes and fibroblasts cells present in TME can effectively induce NF-κB signaling and upon nuclear translocation, NF-κB significantly enhances growth, metastasis and survival of CSCs in colorectal cancer. The TNF-β and TGF-β3 are secreted by T lymphocytes and fibroblasts in inducing NF-κB signaling. Noteworthy, Res (0–10 μM) prevents secretion of TNF-β and TGF-β3 by lymphocytes and fibroblasts in reducing CSC features in colorectal cancer [360]. Another experiment reveals that cancer-associated fibroblasts (CAFs) existing in TME are able to reduce SOX2 expression and suppress stemness in breast cancer via reducing CSC features [361].

Another strategy that can be followed by Res, is the induction of endothelial differentiation. For this purpose, Res and sulindac stimulate vascular endothelial cadherin (VE-cadherin) and von Willebrand factor (vWF) in triggering trans-differentiation in CSCs and mediating their transformation into endothelial lineage [362]. Recently, attention has been directed towards using siRNA and natural compounds in synergistic cancer therapy [312]. An experiment has shown that RAD51 has oncogenic role and enhances CSC features in cervical cancer. A combination of Res and RAD51-siRNA induces apoptosis in HeLa cells and prevents cervical cancer progression [363]. Overall, experiments reveal role of Res in suppressing CSCs in cancer progression (Figure 6, Table 7) [364,365,366,367].

### 5.7. Berberine

The berberine (BBR) is an isoquinoline alkaloid mainly derived from *Coptis chinesis* with great therapeutic impacts [375]. In addition, BBR can be isolated from other *Berberis* plants such as Berberis julianae and Scutellaria baicalensis [376]. Although BBR has pharmacological activities including antioxidant, anti-inflammatory, anti-diabetes, hepatoprotective and renoproective, among others, much attention has been directed towards its anti-tumor activities [377,378,379,380]. The BBR induces apoptosis, autophagy and cycle arrest (G1 phase) in colon cancer cells. Mechanistically, BBR enhances PTEN expression, while it reduces expression of PI3K, Akt and mTOR [151]. Furthermore, BBR suppresses fatty acid metabolism and diminishes extracellular vesicle production in favor of cancer growth suppression [381]. The BBR is able to decrease expression level of IGF2BP3 in triggering cycle arrest (G0/G1 phase) in colorectal cancer cells [382]. The current section emphasizes on BBR function in affecting CSCs.

The pancreatic cancer treatment is an increasing challenge for physicians, as the tumor cells have high growth and migration capabilities [383,384]. Furthermore, they can develop drug resistance and risk of recurrence is present in pancreatic cancer patients. The CSCs play a significant role in aforementioned processes [385,386,387,388]. An experiment has shown that BBR (15 µM) reduces expression levels of SOX2, Nanog and POU5F1 as CSC markers to impair resistance of pancreatic cancer cells to gemcitabine and suppress their progression [389]. One of the targets of BBR in cancer therapy is miRNAs. Overall, the expression level of tumor-promoting miRNAs undergoes down-regulation by BBR, while an increase occurs in expression profile of tumor-suppressor miRNAs [390,391,392]. The oral CSCs have self-renewal capacity and demonstrate high colony formation and migration abilities that are in favor of triggering drug resistance. The miRNA-21 overexpression plays a tumor-promoting role in oral CSCs. The BBR (10 μM) reduces expression level of miRNA-21 to impair self-renewal capacity of CSCs and decrease ALDH1 expression. Then, a significant increase occurs in sensitivity of oral CSCs to cisplatin and 5-flourouracil chemotherapy [393]. 

The glioma-associated oncogene-1 (Gli1) is a tumor-promoting factor that its upregulation enhances growth and invasion of colorectal cancer cells [394]. Noteworthy, Gli1 participates in enhancing CSCs features and mediating tumor progression [395]. Therefore, its inhibition by BBR can exert anti-tumor activities. A recent experiment has shown that chemotherapy enhances ovarian cancer progression via enhancing CSCs features. The Gli1 and its downstream target BMI1 are involved in promoting CSCs features in ovarian cancer. The BBR administration down-regulates Gli1 expression to inhibit BMI1, leading to a decrease in CSC features [396]. Overall, CD133, β-catenin, n-Myc, nestin, SOX2 and Notch2 undergo down-regulation by BBR in suppressing CSC features [397]. Furthermore, BBR co-administration with Dodecyl-TPP (d-TPP) can synergistically suppress CSC features in breast cancer [205]. Moreover, to selectively target CSCs, nanostructures can be utilized [398,399]. In an effort, BBR was loaded on liposomal nanocarriers and results demonstrated potential of these nanocarriers in crossing over CSC membrane. Then, they can reduce expression level of ABCC1, ABCC2, ABCC3 and ABCG2. Furthermore, BBR-loaded liposomes selectively internalize in mitochondria and induce apoptosis via triggering mitochondrial membrane potential loss, enhancing Bax expression and reducing Bcl-2 level. Then, cytochrome C (cyt C) release occurs and promotes caspase-3 and -9 expressions to mediate apoptosis in CSCs and suppress breast cancer progression. Furthermore, in vivo experiment on xenografts in nude mice also has shown potential of BBR-loaded liposomes in suppressing tumor growth (Table 8) [400].

### 5.8. Ginseng and Its Derivatives

The ginsenosides are derivatives from ginseng that have therapeutics impacts including anti-diabetes, anti-inflammatory and neuroprotective, and among them, anti-tumor activity of ginsenosides is of importance [401,402,403]. The ginsenoside Rg3 is able to suppress progression of colorectal tumor cells and enhance their sensitivity to oxaliplatin and 5-flouroruacil chemotherapy. Mechanistically, ginsenoside Rg3 (200 μM) reduces levels of CD24, CD44 and EpCAM to inhibit stemness and population of CSCs [404]. The ginsenoside Rh2 also suppress skin cancer progression in a concentration-dependent manner (0–1 mg/mL) and induces autophagy to suppress β-catenin signaling and impair CSC features in skin cancer. Autophagy inhibition abrogates capacity of ginsenoside Rh2 in suppressing β-catenin signaling and CSC features in skin cancer. Therefore, autophagy induction by ginsenoside Rh2 is of importance for inhibiting β-catenin-mediated CSC features in skin cancer [405]. However, a special attention should be directed towards autophagy, as it has both tumor-promoting and tumor-suppressor roles in cancer [355]. An experiment demonstrates that ginsenoside F2 (0–120 μM) stimulates apoptosis and autophagy in breast CSCs. Noteworthy, autophagy plays as a pro-survival mechanism and autophagy inhibition increases ginsenoside F2 capacity in mediating apoptosis in CSCs [406]. The ginsenoside Rb1 and compound K are able to suppress self-renewal capacity of CSCs in ovarian cancer. Furthermore, Rb1 and compound K enhance sensitivity of ovarian cancer cells to cisplatin and paclitaxel chemotherapy via suppressing CSCs. The drug sensitivity activity of Rb1 and compound K are mediated via suppressing β-catenin signaling and subsequent down-regulation of ABCG2 and P-gp. Furthermore, Rb1 and compound K suppress CSC migration and invasion via EMT inhibition [407]. For suppressing self-renewal capacity of CSCs, ginsenoside Rg3 (0–100 μM) suppresses Akt signaling. Furthermore, ginsenoside Rg3 reduces HIF-1α expression to down-regulate SOX-2 and Bmi-1 expression levels, leading to decreased stemness and CSC features in breast cancer [408]. Overall, ginsenosides similar to genistein, are able to suppress cancer progression via affecting CSCs [409,410,411]. Besides, ginsenosides are derived from ginseng extract and it has been reported that ginseng extract is also capable of suppressing CSC features in cancer (Figure 7, Table 9) [412,413]. Figure 8 and Figure 9 show the chemical structures of selected phytochemicals discussed in the current review.

## 6. Conclusions and Remarks

A variety of factors have been found to be responsible for tumor progression. The CSCs role in tumor progression has been confirmed in different experiments. In addition, molecular pathways participating in CSC feature suppression/induction have been shown. The CSCs enhance stemness of tumor cells and can mediate their growth and migration. Furthermore, CSCs are able to reduce sensitivity of tumor cells to chemotherapy and radiotherapy. The CD44, CD33, ALDH1, Nanog, SOX2 and Oct2 are the most well-known CSC markers. The overexpression of these factors significantly enhances stemness of tumors. Various molecular pathways are able to enhance cancer progression and stemness that include STAT3, NF-κB, Wnt/β-catenin signaling and Sonic Hedgehog, among others. Such molecular pathways were discussed in main text and it was shown that their inhibition can reduce expression of CSC markers and mediate tumor suppression.

Different strategies can be employed for suppressing CSCs. One of the potential methods is application of gene therapy. As signaling networks involved in regulating CSC features have been recognized, gene therapy can inactivate/activate molecular pathways and modulate CSC markers. However, there are some drawbacks with gene therapy, such as high cost. Therefore, plant derived-natural products are of importance due to their multitargeting capacity. The present review attempted to reveal the possible role of selected dietary agents in suppressing CSCs. As it was mentioned in main text and Tables, various doses of these anti-cancer compounds have been utilized and most of them are less than 100 μM. However, each natural product has its own IC_50_ and based on various experiments performed in this case, it is possible to determine optimal dose for using in cancer therapy. One of the limitations of experiments is lack of using nanostructures for delivery of dietary agents and enhancing their capacity in CSC suppression. Hence, future experiments can focus on this aspect and different nanoparticles such as lipid-based nanoparticles, carbon-based nanoparticles and polymeric nanoparticles, among others, can be utilized for targeted delivery. Furthermore, nanostructures overcome poor bioavailability of dietary agents in CSC suppression.

Noteworthy, the molecular pathways inducing CSC marker overexpression, can be targeted by natural products. The PI3K/Akt, Wnt, STAT3, NF-κB and Sonic signaling pathways undergo down-regulation by natural products to reduce expression level of CSC markers including CD44, CD133, Notch, Oct4 and Nanog, impairing cancer stemness and progression. The dietary agents induce apoptosis via both extrinsic and intrinsic pathways to reduce viability of CSCs. These compounds decrease expression level of MMPs and suppress EMT to reduce migration and invasion of CSCs. All these impacts are vital for preventing cancer progression. By suppressing CSC features, natural products enhance sensitivity of tumors to radiotherapy and chemotherapy. However, there are limited experiments in this case. Furthermore, there is lack of study evaluating potential of natural products in targeting CSCs and their interaction with immune system. One of the limitations related that was noted in these studies is that most of them have been performed under in vitro settings and have evaluated the potential impact of dietary agents in targeting CSCs in different cellular models. Hence, additional in vivo studies should be performed to analyze the exact potential of the natural products in targeting CSCs, suppressing tumor progression and investigating possible limitations associated with their use. There are no clinical trials performed so far to evaluate the role of natural products in targeting CSCs in cancer patients. However, there are clinical studies on using dietary agents in treatment of cancer patients (NCT03769766, NCT03980509, NCT01042938; clinicaltrials.gov, accessed on 22 October 2021). Therefore, still enormous work remains to be done to analyze the role of natural products in specifically targeting CSCs and providing effective treatment for cancer patients.

## Figures and Tables

**Figure 1 ijms-22-11669-f001:**
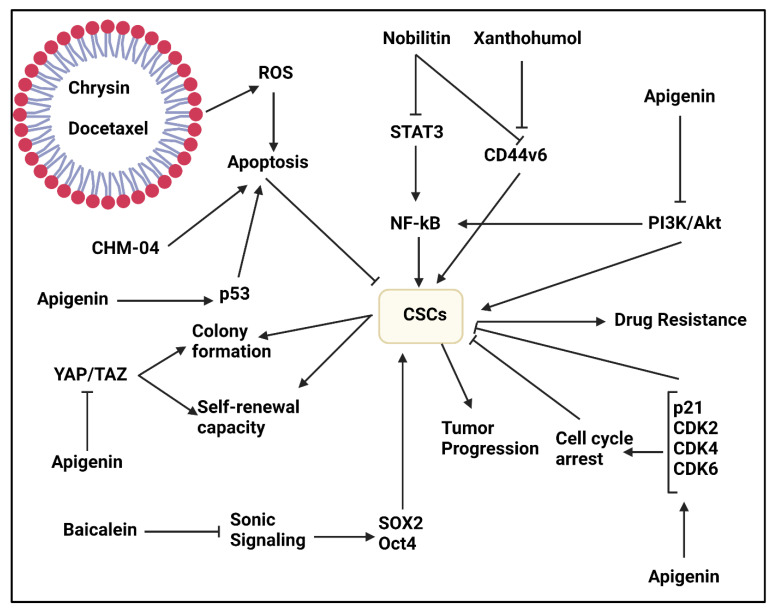
The potential of flavones in suppressing CSCs. Abbreviations: CSCs, cancer stem cells; NF-κB, nuclear factor-kappaB; CDK, cyclin-dependent kinase; Oct4, octamer-4; YAP, Yes-associated protein, SOX2, sex determining region Y-box 2; TAZ, Transcriptional coactivator with PDZ-binding motif; CHM-04, a chrysin derivative; STAT3, signal transducer and activator of transcription 3; Akt, protein kinase-B; PI3K, phosphoinositide 3-kinase; ROS, reactive oxygen species.

**Figure 2 ijms-22-11669-f002:**
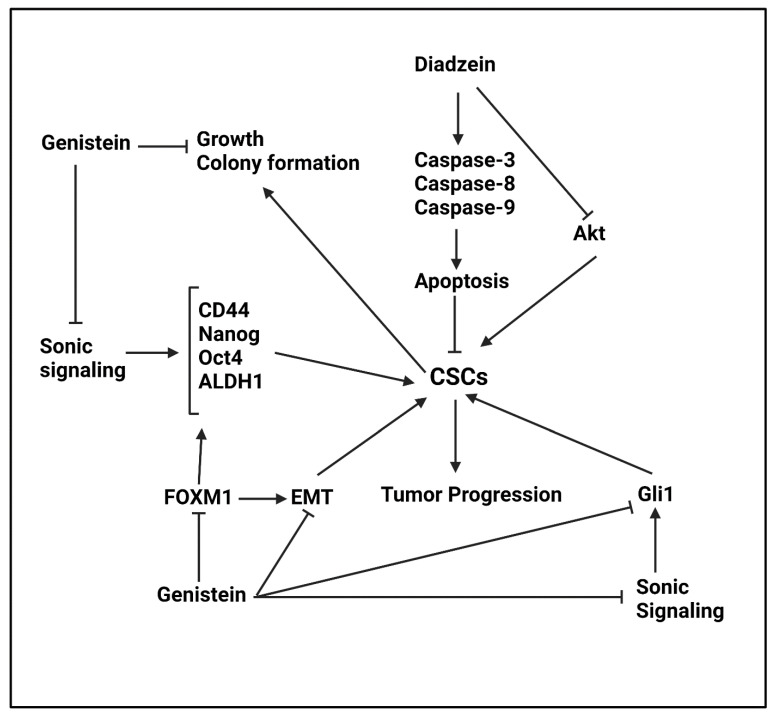
The potential of isoflavonoids in suppressing CSCs.

**Figure 3 ijms-22-11669-f003:**
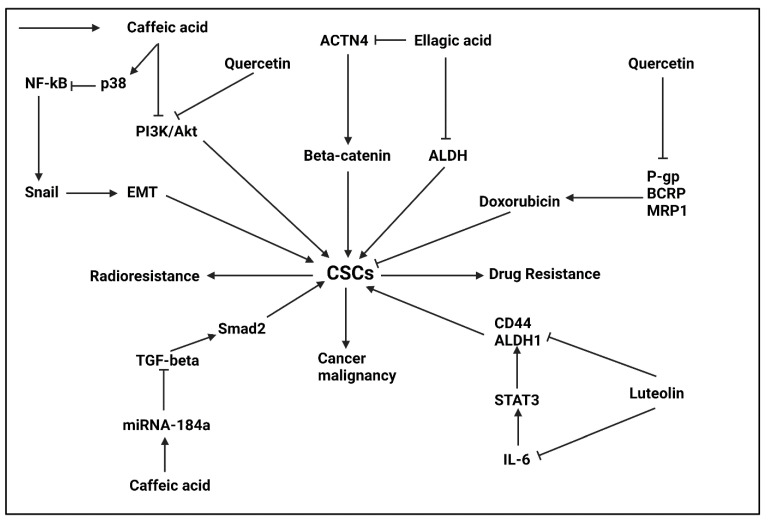
The potential of bioactive constituents of pomegranate in suppressing CSCs.

**Figure 4 ijms-22-11669-f004:**
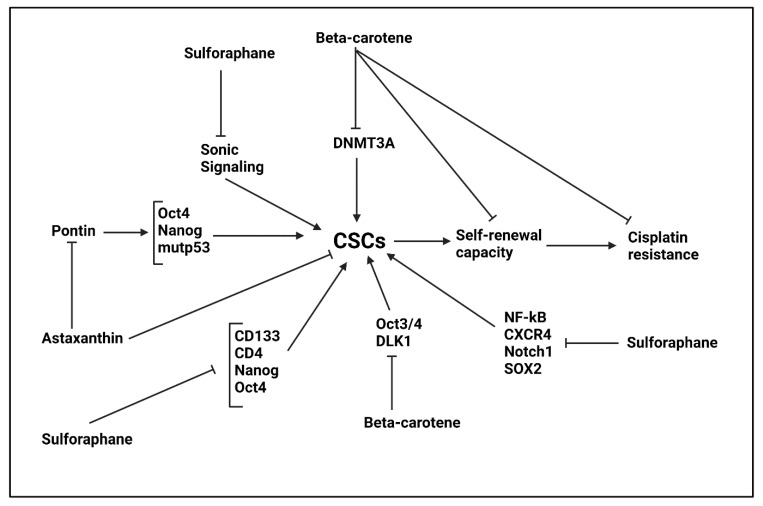
The potential of carotenoids in suppressing CSCs.

**Figure 5 ijms-22-11669-f005:**
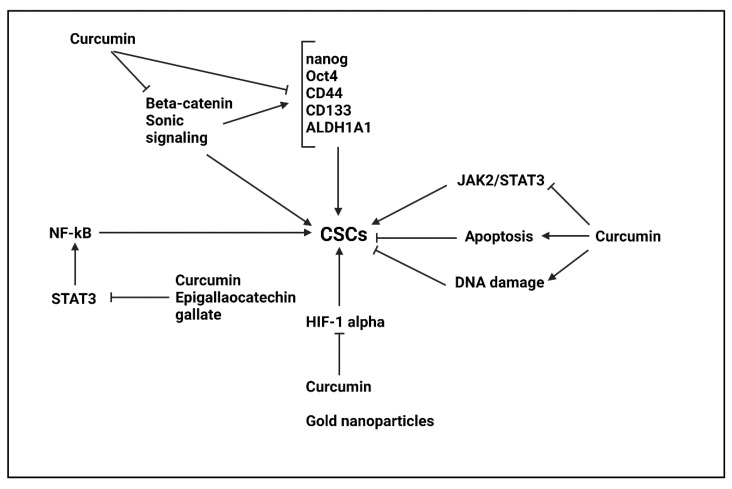
Curcumin in inhibiting CSCs.

**Figure 6 ijms-22-11669-f006:**
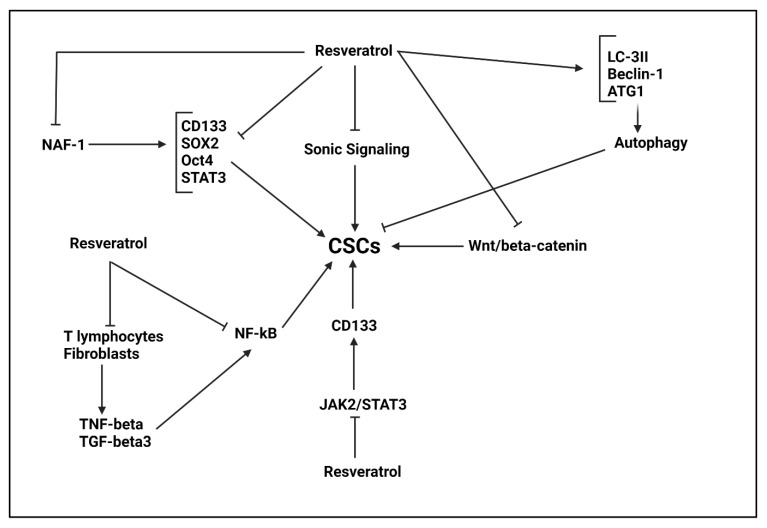
Res in suppressing CSCs in tumor therapy.

**Figure 7 ijms-22-11669-f007:**
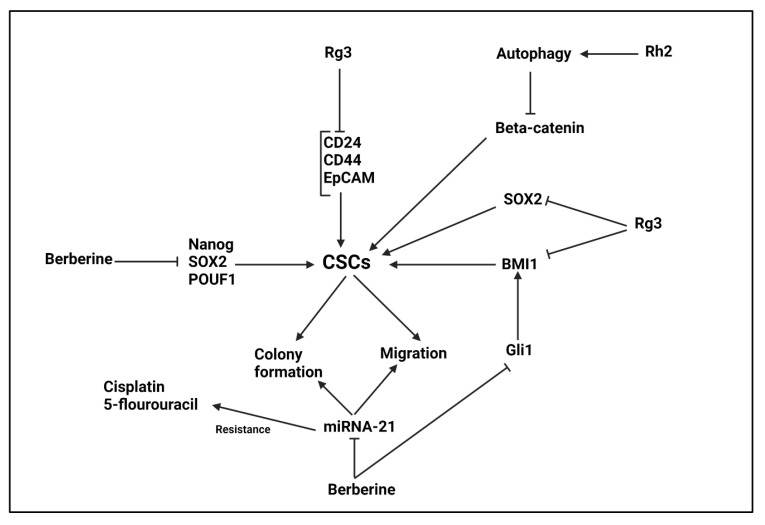
The berberine and ginsenosides as modulators of CSCs in tumor eradication.

**Figure 8 ijms-22-11669-f008:**
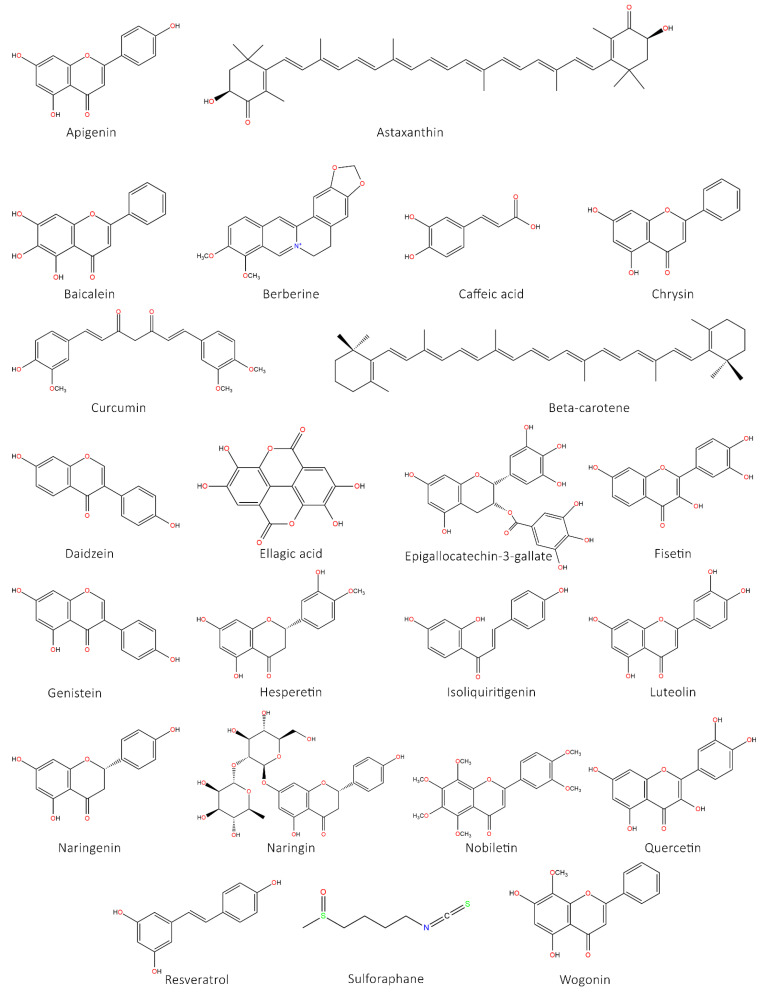
The chemical structures of selected phytochemicals discussed in the review.

**Figure 9 ijms-22-11669-f009:**
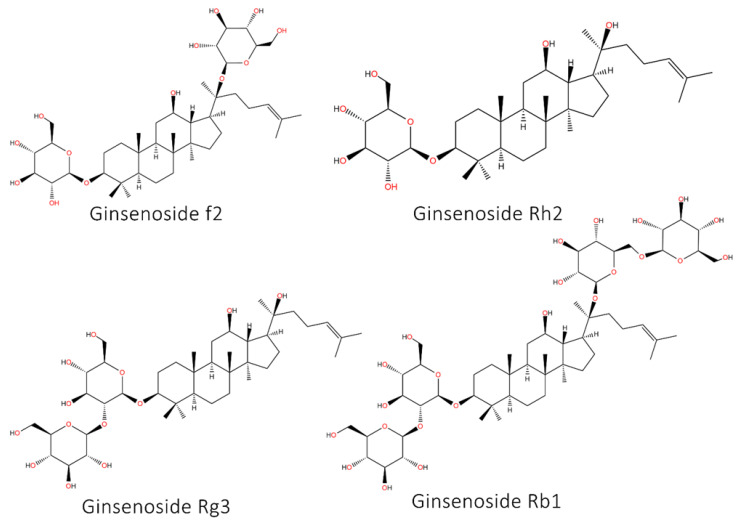
Chemical structures of ginsenosides.

**Table 1 ijms-22-11669-t001:** Flavones in targeting CSCs for tumor suppression.

Anti-Tumor Agent	Cancer Type	In Vitro/In Vivo	Cell Line/Animal Model	Study Design	Remarks	Refs
Chrysin	Colorectal cancer	In vitro	HT-29 cells	-	The chrysin- and docetaxel-loaded micelles exert synergistic therapy in suppressing growth and invasion of CSCs Enhancing ROS production to mediate cell death	[129]
CHM-04 (chrysin derivative)	Breast cancer	In vitro	MCF-7 and MDA-MB-231 cells	10 μM	Higher anti-tumor potential (3.2-fold increase) compared to chrysin Inducing apoptosis and reducing migration Suppressing colony formation of CSCs	[130]
Apigenin	Breast cancer	In vitro In vivo	MDA-MB-231 cells Nude mice	0–64 μM	Suppressing CSC features in breast cancer via inhibiting YAP/TAZ axis	[134]
Apigenin	Lung cancer	In vitro	A549 and H1299 cells	10, 20 and 30 μM	Enhancing p53 expression and impairing CSC features Promoting sensitivity of cancer cells to cisplatin chemotherapy	[151]
Apigenin	Prostate cancer	In vitro	PC3 cells	25 μM	Reducing viability of CSCs and triggering apoptosis via p21 and p27 upregulation Triggering extrinsic pathway of apoptosis Suppressing PI3K/Akt/NF-κB axis	[136]
Apigenin	Prostate cancer	In vitro	PC3 cells	15 μM	Inducing apoptosis and cell cycle arrest in CSCs Suppressing migration and invasion of CSCs Potentiating anti-tumor activity of cisplatin Inhibiting PI3K/Akt and NF-κB pathways	[137]
Apigenin	Head and neck cancer	In vitro	HN-8, HN-30, and HSC-3 cells	0–100 μM	Decreasing viability of cancer cells in a dose-dependent manner Suppressing CSC features via reducing expression levels of CD44, Nanog and CD105	[138]
Apigenin	Prostate cancer	In vitro	PC3, LNCaP, or isolated CD44+ CD133+ and CD44+ stem cells	0–2 μM	Inducing apoptosis in CSCs Suppressing proliferation and viability of CSCs Upregulating expression levels of p21, p27, Bax, Bid, caspase-3 and caspase-8 Inhibiting ERK, PARP and NF-κB expression levels	[139]
Baicalein	Pancreatic cancer	In vitro	PANC-1, BxPC-3, and SW1990 cells	0–300 μM	Reducing SOX-2 expression and impairing CSC features via down-regulating Sonic expression	[144]
Wogonin	Osteosarcoma	In vitro	CD133+ Cal72 cells	0–80 μM	Triggering cell death in cancer cells via enhancing ROS levels	[149]
Wogonin	Osteosarcoma	In vitro	CD133+ CAL72 cells	0–80 μM	Triggering apoptosis in CSCs Inhibiting self-renewal capacity Reducing migration and invasion of CSCs via reducing MMP-9 expression	[150]

**Table 2 ijms-22-11669-t002:** Gallate in suppressing CSCs.

Cancer Type	In Vitro/In Vivo	Cell Line/Animal Model	Study Design	Remarks	Refs
Lung cancer	In vitro In vivo	A549 and H1299 cell lines Xenograft model	0–40 μM 20 mg/kg	Suppressing self-renewal capacity of tumor cells CLOCK down-regulation Suppressing tumor growth in vivo	[163]
Lung cancer	In vitro	A549, H460, H1299, and HEK-293T cells	0–40 μM	Suppressing CSC features in lung cancer Promoting miRNA-485-5p expression Down-regulating PXRα expression	[168]
Lung cancer	In vitro	A549 and H1299 cells	0–100 μM	The enrichment of CSC features in lung cancer Wnt/β-catenin inhibition prevents CSC characteristics	[169]
Colorectal cancer	In vitro	DLD-1 and SW480 cells	0–60 μM	Proliferation inhibition Apoptosis induction Impairing CSC features via inhibiting Wnt/β-catenin axis	[170]
Colorectal cancer	In vitro	HCT116 cells	0–200 μM	Inhibiting self-renewal capacity of CSCs Reducing expression levels of Notch1, Suz12 and EZH2 Increasing sensitivity of tumor cells to 5-flourouracil chemotherapy	[173]
Bladder cancer	In vitro	EJ and UM-UC-3 cells	0–90 μM	Inhibiting Sonic Hedgehog signaling Reducing CSC features Apoptosis induction in CSCs Impairing proliferation of CSCs	[174]
Head and neck cancer	In vitro In vivo	K3, K4 and K5 cells Xenograft model	0–10 μM	Impairing CSC features via suppressing Notch signaling	[176]
Nasopharyngeal cancer	In vitro In vivo	CNE2 and C666-1 cells Xenograft model	0–50 μM	Inhibiting self-renewal capacity of CSCs Suppressing migration via EMT inhibition Down-regulating NF-κB expression	[177]

**Table 3 ijms-22-11669-t003:** The potential of genistein in cancer suppression via targeting CSCs.

Cancer Type	In Vitro/In Vivo	Cell Line/Animal Model	Study Design	Remarks	Refs
Nasopharyngeal cancer	In vitro	CNE2 and HONE1 cells	0–100 μM	Reducing expression level of CSC markers including CD44 and CD133 via suppressing Sonic signaling	[193]
Renal cancer	In vitro	786-O and ACHN cell lines	0–90 μM	Suppressing growth and colony formation capacities of CSCs Triggering apoptosis and reducing CSC marker expression Inhibiting Sonic Hedgehog signaling	[205]
Breast cancer	In vitro	MCF-7 and MDA-MB-231 cells	2 μM or 40 nM	Decreasing number of CSCs Inducing PI3K/Akt and MEK/ERK pathways in a paracrine manner and mediating differentiation of CSCs	[195]
Gastric cancer	In vitro	MGC-803 and SGC-7901 cells	15 μM	Suppressing colony formation and self-renewal capacities Inhibiting chemoresistance via down-regulating ERK1/2 and ABCG2	[196]
Gastric cancer	In vitro	SGC-7901 cells	0–10 μmol/L	Decreasing FoxM1 expression to prevent self-renewal capacity and migration of CSCs Decreasing expression levels of CSC markers including CD44, CD133 and ALDH1	[197]
Gastric cancer	In vitro	AGS and MKN45 cells	10 μg/ml	Decreasing expression level of CD44 via suppressing Sonic Hedgehog signaling	[198]
Prostate cancer	In vitro In vivo	22RV1, DU145 cells Xenograft animal model	15 and 30 μM 10 mg/kg	Suppressing Hedgehog/Gli1 axis to impair stemness and CSC features in prostate cancer	[199]

**Table 4 ijms-22-11669-t004:** The CSCs as promising targets of quercetin in tumor therapy.

Cancer Type	In Vitro/In Vivo	Cell Line/Animal Model	Study Design	Remarks	Refs
Breast cancer	In vitro In vivo	MCF-7 cells Nude mice	0–200 μM	Reducing number of CSCs Impairing CSC features Suppressing PI3K/Akt/mTOR axis	[233]
Breast cancer	In vitro	MDA-MB-231 cells	0–200 μM	Disrupting tumor progression and CSC features via down-regulating ALDH1A1, CXCR4, MUC1 and EpCAM	[234]
Breast cancer	In vitro	MCF-7 cells	0–70 μM	Suppressing CSCs in breast cancer via preventing nuclear translocation of YB-1	[235]
Breast cancer	In vitro	MCF-10A, MCF-7, MDA-MB-231 and AC16 cells	0–2 μM	Enhancing intracellular accumulation of doxorubicin via down-regulating P-gp, MRP1 and BCRP Eliminating CSCs and potentiating doxorubicin’s anti-tumor activity	[236]
Pancreatic cancer	In vitro	AsPC1 and PANC1 cells	50 μM	Suppressing self-renewal capacity of tumor cells via enhancing miRNA-200b expression	[238]
Pancreatic cancer	In vitro In vivo	BxPc-3 and MIA-PaCa2 cells Nude mice and xenografts	100, 200 and 400 μM 50 mg/kg	Synergistic impact between quercetin and sulforaphane Down-regulating NF-κB signaling Impairing tumor growth in vitro and in vivo	[240]
Colon cancer	In vitro	DLD-1 and HT-29 cells	0–50 μM	Suppressing Notch signaling, impairing CSCs features and increasing sensitivity of tumor cells to radiotherapy	[241]
Colorectal cancer	In vitro	HT29 cells	0–100 μM	Triggering apoptosis and cell cycle arrest in CSCs Promoting sensitivity of tumor cells to doxorubicin chemotherapy	[242]
Prostate cancer	In vitro	PC3, LNCaP and ARPE-19 cells	40 μM	Queretin and midkine-siRNA co-application suppresses CSC features via dual inhibition of PI3K/Akt and MAPK/ERK molecular pathways	[248]

**Table 5 ijms-22-11669-t005:** The sulforaphane as a potential anti-tumor agent in cancer therapy.

Cancer Type	In vitro/In Vivo	Cell Line/Animal Model	Study Design	Remarks	Refs
Triple-negative breast cancer	In vitro	SUM149 and SUM159 cells	2.5 and 5 μM	Inducing apoptosis in CSCs and reducing ALDH expression as CSC marker	[286]
Melanoma	In vitro In vivo	A375 cells Xenograft	20 μM 10 μM/kg	The EZH2 promotes CSC features in melanoma and is suppressed by sulforaphane	[287]
Pancreatic cancer	In vivo	Mice	20 mg/kg	Suppressing Sonic/Gli1 axis to impair CSC features and self-renewal capacity in pancreatic cancer	[288]
Glioblastoma	In vitro In vivo	U87, U373, U118, and SF767 cells Mice	0–50 μM 100 mg/kg	Suppressing stem cell-like spheroids Inducing apoptosis and eliminating CSCs Promoting drug sensitivity	[289]
Leukemia	In vitro	KU812 cells	0–30 μM	Enhancing ROS levels Suppressing β-catenin signaling to reduce GSH levels Promoting potential of imatinib in tumor suppression	[290]
Lung cancer	In vitro	NSCLC PC9 cells	0–12 μM	Suppressing Sonic Hedgehog signaling Decreasing levels of CD133 and CD44 Inhibiting gefitinib resistance	[291]
Lung cancer	In vitro In vivo	A549 cells Nude mice	0–40 μM 25 and 50 mg/kg	Preventing tobacco-mediated CSC feature acquisition via suppressing IL-6/ΔNp63α/Notch axis	[194]
Oral cancer	In vitro	SAS or GNM cells	0–50 μM	Upregulating miRNA-200c expression to impair CSC features and stemness in oral cancer	[292]
Epidermal squamous cell carcinoma	In vitro	SCC-13 cells	0–20 μM	The SCC-3 cells derived from CSCs have high sensitivity to a combination of sulforaphane and cisplatin This combination reduces viability of tumor cells and their capacity in colony formation	[293]

**Table 6 ijms-22-11669-t006:** Curcumin potential of suppressing CSCs in tumor treatment.

Cancer Type	In Vitro/In Vivo	Cell Line/Animal Model	Study Design	Remarks	Refs
Lung cancer	In vitro	A549 and H1299 cells	0–40 μM	Inducing apoptosis Reducing levels of CSC markers Inhibiting Wnt/β-catenin and Sonic signaling pathways	[303]
Bladder cancer	In vitro	UM-UC-3 and EJ cells	0–50 μM	Apoptosis induction Decreasing expression levels of CD44, CD133, Oct4, Nanog and ALDH1 Inhibiting Sonic Hedgehog signaling	[320]
Breast cancer	In vitro	MCF-10A and MCF-7 cells	8 μM	A combination of curcumin and quinacrine induces DNA damage and reduces ABCG2 expression to impair cancer progression	[322]
Breast cancer	In vitro	SUM159 and MCF7 cell lines	0–40 μM	Reducing colony formation capacity of CSCs Down-regulating expression levels of CD44, ALDH1, Nanog and Oct4 Inhibiting Wnt/β-catenin axis	[333]
Breast cancer	In vitro	MDA-MB-231 and MCF-7 cells	5 μmol/L	Inducing apoptosis via Bcl-2 down-regulation Enhancing sensitivity of breast CSCs to mitomycin C	[328]
Brain cancer	In vitro	U87MG cells	-	The curcumin-loaded nanoparticles efficiently penetrate into BBB to induce apoptosis in CSCs and reduce tumor progression	[332]
Pancreatic cancer	In vitro In vivo	BxPC3, MiaPaCa2 and Panc1 PDAC cells Nude mice	0–20 μM 100 mg/kg	Suppressing self-renewal capacity of tumor cells Retarding tumor growth in vivo Enhancing sensitivity to gemcitabine chemotherapy	[339]
Prostate cancer	In vitro	DU145 cells	-	Inhibiting growth and invasion of prostate CSCs Overexpression of miRNA-770-5p and miRNA-1247	[337]
Colorectal cancer	In vitro	HCT116 and DLD1 cells	0–20 μM	Reducing expression level of CSC markers including CD44, Oct4 and ALDH1 Inhibiting STAT3 signaling	[338]

**Table 7 ijms-22-11669-t007:** Determining resveratrol potential in targeting CSCs and suppressing tumor progression.

Cancer Type	In Vitro/In Vivo	Cell Line/Animal Model	Study Design	Remarks	Refs
Pancreatic cancer	In vitro	MiaPaCa-2 and Panc-1 cells	50 μmol/L	Promoting gemcitabine sensitivity Suppressing lipid metabolism Down-regulation of SREBP1 Inhibiting stemness Suppressing colony formation and CSC features	[368]
Pancreatic cancer	In vitro In vivo	CSCs and mice	0–30 μM	Inhibiting tumor growth and development in vivo Inducing apoptosis in CSCs via upregulating caspase-3 and -7, and down-regulating Bcl-2 and XIAP Decreasing expression levels of Nanog, SOX2, c-Myc and Oct4 as CSC markers Reducing ABCG2 expression as an inducer of drug resistance in CSCs	[369]
Medulloblastoma	In vitro	CSCs derived from medulloblastoma	150 μM	Suppressing proliferation of CSCs and increasing sensitivity to radiotherapy	[370]
Ovarian cancer	In vitro	A2780 cells	0–50 μM	Suppressing self-renewal capacity of CSCs Triggering cell death in CSCs via mediating ROS overgeneration	[371]
Breast cancer	In vitro	MCF-7 cells	0–500 μM	Preventing proliferation of CSCs Triggering apoptosis Mediating oxidative damage Caspase cascade activation and inducing PARP cleavage Reducing SOD, MnSOD and catalase levels	[372]
Nasopharyngeal carcinoma	In vitro	TW01, TW06, and HONE-1 cells	0–100 μM	Suppressing self-renewal capacity and migration of CSCs P53 overexpression EMT inhibition	[373]
Colorectal cancer	In vitro	HCT116 cells	5 μM	Upregulation of CD133, CD44 and ALDH1 as CSC markers by TGF-β Suppressing CSC features by resveratrol and triggering apoptosis in CSCs by caspase-3 upregulation Promoting drug sensitivity	[374]

**Table 8 ijms-22-11669-t008:** The BBR in targeting CSCs for tumor suppression.

Cancer Type	In Vitro/In Vivo	Cell Line/Animal Model	Study Design	Remarks	Refs
Pancreatic cancer	In vitro	PANC-1 and MIA PaCa-2 cells	15 μM	Reducing population of CSCs Decreasing expression level of CSC markers including SOX2, ALDH1, Nanog and POU5F1 Promoting sensitivity of tumor cells to gemcitabine chemotherapy	[389]
Oral cancer	In vitro	SAS and OECM-1 cells	0–40 μM	Suppressing tumor progression in a dose-dependent manner Reducing colony formation, migration and self-renewal capacity of tumor cells Reducing miRNA-21 expression to impair CSC features	[393]
Ovarian cancer	In vitro	SKOV3 and A2780 cells	5 μM	Chemotherapy promotes Gli1 expression and facilitates CSC features Suppressing Gli1 pathway by berberine Inhibiting EMT-mediated metastasis and reducing CSC features	[396]
Neuroblastoma	In vitro	N2a cells	10 and 20 μg/mL	Inducing apoptosis and cell cycle arrest in tumor cells Suppressing metastasis via EMT inhibition Inhibiting CSCs features	[397]
Breast cancer	In vitro In vivo	MCF-7 cells Xenografts	40 μM of liposomal berberine	The berberine-loaded liposomes selectively target CSCs and induce apoptosis Enhancing berberine accumulation in tumor cells Retarding tumor growth in vivo	[400]

**Table 9 ijms-22-11669-t009:** Ginsenosides as potential inhibitors of CSCs in different tumors.

Anti-Tumor Agent	Cancer Type	In Vitro/In Vivo	Cell Line/Animal Model	Study Design	Remarks	Refs
Ginsenoside Rg3	Colorectal cancer	In vitro In vivo	LoVo, SW620 and HCT116 cells Nude mice	-	Promoting anti-tumor potential of oxaliplatin and 5-flourouracil Suppressing stemness and CSC features	[404]
Ginsenoside Rh2	Skin cancer	In vitro	A431 cells	0–1 mg/mL	Inhibiting viability of tumor cells Autophagy induction and β-catenin signaling inhibition	[405]
Ginsenoside Rb1	Ovarian cancer	In vitro In vivo	SKOV-3 and HEYA8 cells Mice	0–500 nM 50 mg/kg	Suppressing Wnt signaling and reversing EMT to impair cancer stemness and inhibit drug resistance	[407]
Gisnenoside Rg3	Breast cancer	In vitro	MCF-7 and MDA-MB-231 cells	0–100 μM	Inhibiting Akt/HIF-1α axis Reducing expression levels of SOX2 and Bmi-1 Inhibiting CSC features and self-renewal capacities	[408]
Ginsenoside F2	Breast cancer	In vitro	MCF-7 cells	0–140 μM	Apoptosis induction in CSCs Inhibiting autophagy promotes potential of ginsenoside F2 in tumor suppression	[406]
Ginsenoside Rh2	Hepatocellular carcinoma	In vitro In vivo	HepG2 and Huh7 cells Mice	0–1 mg/mL 1 mg/kg	Inhibiting CSC features in a dose-dependent manner Autophagy induction and suppressing β-catenin signaling	[409]

## Data Availability

Not applicable.

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
