# Peer review of "Targeting Cancer Stem Cells by Dietary Agents: An Important Therapeutic Strategy against Human Malignancies"

_ijms, 2021, doi:10.3390/ijms222111669_

Round 1
Reviewer 1 Report
It is a well-written review paper.
Natural products derived from plants, including dietary agents, have been used in cancer therapies for a long time. This review focuses on CSC and dietary agents and it lists many data and they are useful.
Most experimental data were obtained from in vitro experiments. Can you please give a reason or give some insights on this issue, i.e. lack of in vivo data?
It seems systematic approaches if reported, e.g. high throughput screening of dietary agents (libraries), were not mentioned in this review. Those data are very useful resources. Please add those information if available.
Author Response
Comment: It is a well-written review paper. Natural products derived from plants, including dietary agents, have been used in cancer therapies for a long time. This review focuses on CSC and dietary agents and it lists many data and they are useful.
Response: We really appreciate you for providing valuable comments on our manuscript.
Comment: Most experimental data were obtained from in vitro experiments. Can you please give a reason or give some insights on this issue, i.e. lack of in vivo data?
Response: We really appreciate your comment. Yes, you are true. Most of the experiments are in vitro. However, there are also in vivo experiments that we have listed in tables and in main text. As majority of experiments are in vitro, we have discussed this aspect in depth in the conclusion section.
Comment: It seems systematic approaches if reported, e.g. high throughput screening of dietary agents (libraries), were not mentioned in this review. Those data are very useful resources. Please add those information if available.
Response: We really appreciate your comment. As impact of dietary agents on CSCs is a new field for cancer suppression, there were many natural products that their potential in affecting CSCs has not been investigated yet. For choosing other natural products, we have first searched classes as natural products such as flavonoids and then, examined role of selected compounds under these specific classes in affecting CSCs. However, we have made no libraries. For more clarity, we have added a section about our search criteria.
Reviewer 2 Report
This is a very interesting and comprehensive review about the role of dietary bioactive compounds in targeting cancer stem cells as a novel therapeutic strategy against human malignancies.
The Review is well written and many tables and/or charts are shown to help the reader in better and more straightforwardly catching the content of what in the text.
If authors agree, I would just suggest them to add a couple of recent articles that they might ignore but that could be helpful to add.
In particular: "Intorduction": lines 58-59: PMID: 34199263 and lines 72-75: PMID: 31428230
Author Response
Reviewer#2
This is a very interesting and comprehensive review about the role of dietary bioactive compounds in targeting cancer stem cells as a novel therapeutic strategy against human malignancies.
Comment: The Review is well written and many tables and/or charts are shown to help the reader in better and more straightforwardly catching the content of what in the text.
Response: Thanks for your useful comments about our article.
Comment: If authors agree, I would just suggest them to add a couple of recent articles that they might ignore but that could be helpful to add.
In particular: "Introduction": lines 58-59: PMID: 34199263 and lines 72-75: PMID: 31428230
Response: Thanks for your suggestions. They were helpful and we have added them.